# INTENTION-CONDITIONED FLOW OCCUPANCY MODELS

**Chongyi Zheng**[1]    **Seohong Park**[2]    **Sergey Levine**[2]    **Benjamin Eysenbach**[1]

[1]Princeton University        [2]University of California, Berkeley

chongyiz@princeton.edu

## ABSTRACT

Large-scale pre-training has fundamentally changed how machine learning research is done today: large foundation models are trained once, and then can be used by anyone in the community (including those without data or compute resources to train a model from scratch) to adapt and fine-tune to specific tasks. Applying this same framework to reinforcement learning (RL) is appealing because it offers compelling avenues for addressing core challenges in RL, including sample efficiency and robustness. However, there remains a fundamental challenge to pre-train large models in the context of RL: actions have long-term dependencies, so training a foundation model that reasons across *time* is important. Recent advances in generative AI have provided new tools for modeling highly complex distributions. In this paper, we build a probabilistic model to predict which states an agent will visit in the temporally distant future (i.e., an occupancy measure) using flow matching. As large datasets are often constructed by many distinct users performing distinct tasks, we include in our model a latent variable capturing the user's intention. This intention increases the expressivity of our model and enables adaptation with generalized policy improvement. We call our proposed method **intention-conditioned flow occupancy models (InFOM)**. Comparing with alternative methods for pre-training, our experiments on 36 state-based and 4 image-based benchmark tasks demonstrate that the proposed method achieves $1.8\times$ median improvement in returns and increases success rates by $36\%$.

Website: https://chongyi-zheng.github.io/infom

Code: https://github.com/chongyi-zheng/infom

## 1 INTRODUCTION

Many of the recent celebrated successes of machine learning have been enabled by training large foundation models on vast datasets, and then adapting those models to downstream tasks. Examples include today's chatbots (e.g., Gemini (Team et al., 2023) and ChatGPT (Achiam et al., 2023)) and generalist robotic systems (e.g., $\pi_0$ (Black et al., 2024) and Octo (Team et al., 2024)). This pre-training-fine-tuning paradigm has been wildly successful in fields ranging from computer vision to natural language processing (Devlin et al., 2019; Brown et al., 2020; Touvron et al., 2023; Zhai et al., 2023; Radford et al., 2021; He et al., 2022; Ouyang et al., 2022; Lu et al., 2019), yet harnessing it in the context of reinforcement learning (RL) remains an open

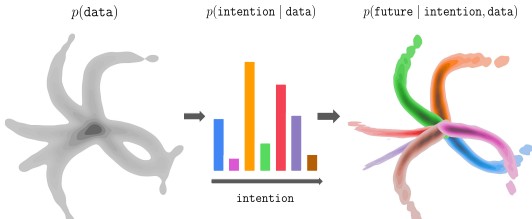

$p(\text{data})$     $p(\text{intention} \mid \text{data})$     $p(\text{future} \mid \text{intention}, \text{data})$

intention

Figure 1: **InFOM** is a latent variable model for pre-training and fine-tuning in reinforcement learning. *(Left)* The datasets are collected by users performing distinct tasks. *(Center)* We encode intentions by maximizing an evidence lower bound of data likelihood, *(Right)* enabling intention-aware future prediction using flow matching. See Sec. 4 for details.

problem. What fundamentally makes the RL problem difficult is reasoning about time and intention—an effective RL agent must reason about the long-term effect of actions taken now, and must recognize that the data observed are often collected by distinct users performing multiple tasks. However, current attempts to build foundation models for RL often neglect these two important bits, focusing

on predicting the actions in the pre-training dataset instead (Team et al., 2024; O'Neill et al., 2024; Walke et al., 2023).

The closest attempts to building RL algorithms that capture temporal bits are those based on world models (Ding et al., 2024; Hafner et al., 2023; Mendonca et al., 2021) and those based on occupancy models (Janner et al., 2020; Blier et al., 2021; Zheng et al., 2024b; Farebrother et al., 2025).[1] World models can achieve great performance in sample efficiency (Janner et al., 2019) and generalize to diverse tasks (Hafner et al., 2023; Mendonca et al., 2021), although their capacity to perform long-horizon reasoning remains limited because of compounding errors (Talvitie, 2014; Janner et al., 2019; Lambert et al., 2022). Occupancy models (Dayan, 1993) and variants that enable scaling to high-dimensional tasks can also achieve great performance in predicting future events (Sikchi et al., 2024; Barreto et al., 2018; Zheng et al., 2024b; 2025; Farebrother et al., 2025), but are typically hard to train and ignore user intentions. Recent advances in generative AI (e.g., flow-matching (Lipman et al., 2024; 2023; Liu et al., 2023) and diffusion (Ho et al., 2020; Song et al., 2021) models) enable modeling complex distributions taking various inputs, providing new tools for constructing occupancy models that depend on intentions.

In this paper, we propose a framework (Fig. 1) for pre-training in RL that simultaneously learns a probabilistic model to capture bits about time and intention. Building upon prior work on variational inference (Kingma & Welling, 2013; Alemi et al., 2017) and successor representations (Janner et al., 2020; Touati & Ollivier, 2021; Barreto et al., 2017; Zheng et al., 2024b; Farebrother et al., 2025), we learn latent variable models of temporally distant future states, enabling intention-aware prediction. Building upon prior work on generative modeling, we use an expressive flow matching method (Farebrother et al., 2025) to train occupancy models, enabling highly flexible modeling of occupancy measures. We call the resulting algorithm **intention-conditioned flow occupancy models (InFOM)**. Experiments on 36 state-based and 4 image-based benchmark tasks show that InFOM outperforms alternative methods for pre-training and fine-tuning by $1.8\times$ median improvement in returns and $36\%$ improvement in success rates. Additional experiments demonstrate that our latent variable model is capable of inferring underlying user intentions (Sec. 5.2) and enables efficient policy extraction (Sec. 5.3).

## 2 Related Work

**Offline unsupervised RL.** The goal of offline unsupervised RL is to pre-train policies, value functions, or models from an unlabeled (reward-free) dataset to enable efficient learning of downstream tasks. Prior work has proposed diverse offline unsupervised RL approaches based on unsupervised skill learning (Touati & Ollivier, 2021; Frans et al., 2024; Park et al., 2024b; Kim et al., 2024; Hu et al., 2023), offline goal-conditioned RL (Eysenbach et al., 2019; 2022; Valieva & Banerjee, 2024; Park et al., 2023a; Zheng et al., 2024b; Park et al., 2025a), and model-based RL (Mendonca et al., 2021; Mazzaglia et al., 2022). Among these categories, our method is conceptually related to offline unsupervised skill learning approaches (Park et al., 2024b; Touati et al., 2023), which also learns a model that predicts intentions. However, our approach differs in that it does not learn multiple skills during pre-training. Our work is complementary to a large body of prior work on using behavioral cloning for pretraining (O'Neill et al., 2024; Team et al., 2024), demonstrating that there are significant additional gains in performance that can be achieved by modeling intentions and occupancy measures simultaneously.

**Unsupervised representation learning for RL.** Another way to leverage an unlabeled offline dataset is to learn representations that facilitate subsequent downstream task learning. Some works adapt existing representation learning techniques from computer vision, such as contrastive learning (He et al., 2020; Parisi et al., 2022; Nair et al., 2023) and masked autoencoding (He et al., 2022; Xiao et al., 2022). Others design specific methods for RL, including self-predictive representations (Schwarzer et al., 2020; Ni et al., 2024) and temporal distance learning (Sermanet et al., 2018; Ma et al., 2023; Mazoure et al., 2023). Those learned representations are typically used as inputs for policy and value networks. The key challenge with these representation learning methods is that it is often (Laskin et al., 2020), though not always (Zhang et al., 2021), unclear whether the learned

---

[1]We will use "successor representations," "occupancy measures," and "occupancy models" interchangeably.

representations will facilitate policy adaptation. In our experiments, we demonstrate that learning occupancy models enables faster policy learning.

**RL with generative models.** Modern generative models have been widely adopted to solve RL problems. Prior work has employed autoregressive models (Vaswani et al., 2017), iterative generative models (e.g., denoising diffusion (Sohl-Dickstein et al., 2015; Ho et al., 2020) and flow matching (Liu et al., 2023; Lipman et al., 2023; 2024)), or autoencoders (Kingma & Welling, 2013) to model trajectories (Chen et al., 2021; Janner et al., 2021; 2022; Ajay et al., 2023), environment dynamics (Ding et al., 2024; Alonso et al., 2024), skills (Ajay et al., 2021; Pertsch et al., 2021; Frans et al., 2024), policies (Wang et al., 2023; Hansen-Estruch et al., 2023; Park et al., 2025b), and values (Dong et al., 2025; Agrawalla et al., 2025). We employ a state-of-the-art flow-matching objective (Farebrother et al., 2025) to model discounted state occupancy measures.

**Successor representations and successor features.** Prior work has used successor representations (Dayan, 1993) and successor features (Barreto et al., 2017) for transfer learning (Barreto et al., 2017; 2018; Borsa et al., 2018; Nemecek & Parr, 2021; Kim et al., 2022), unsupervised RL (Machado et al., 2017; Hansen et al., 2019; Ghosh et al., 2023; Touati et al., 2023; Park et al., 2024b; 2023b; Chen et al., 2023; Zheng et al., 2025; Jain et al., 2023; Zhu et al., 2024), and goal-conditioned RL (Eysenbach et al., 2020; 2022; Zheng et al., 2024b;a). Our method is closely related to prior methods that learn successor representations with generative models (Janner et al., 2020; Thakoor et al., 2022; Tomar et al., 2024; Farebrother et al., 2025). In particular, the most closely related to ours is the prior work by Farebrother et al. (2025), which also uses flow-matching to model the occupancy measures and partly employs the generalized policy improvement (GPI) for policy extraction. Unlike Farebrother et al. (2025), which uses forward-backward representations to capture behavioral intentions and perform GPI over a finite set of intentions, our method employs a latent variable model to learn intentions (Sec. 4.2) and uses an expectile loss to perform implicit GPI (Sec. 4.4). We empirically show that these choices lead to higher returns and success rates (Sec. 5.1, Sec. 5.3).

## 3 PRELIMINARIES

We consider a Markov decision process (MDP) (Sutton et al., 1998) defined by a state space $\mathcal{S}$, an action space $\mathcal{A}$, an initial state distribution $\rho \in \Delta(\mathcal{S})$, a reward function $r : \mathcal{S} \to \mathbb{R}$, a discount factor $\gamma \in [0, 1)$, and a transition distribution $p : \mathcal{S} \times \mathcal{A} \to \Delta(\mathcal{S})$, where $\Delta(\cdot)$ denotes the set of all possible probability distributions over a space. We will use $h$ to denote a time step in the MDP and assume the reward function only depends on the state at the current time step $r_h \triangleq r(s_h)$ without loss of generality (Tomar et al., 2024; Frans et al., 2024; Thakoor et al., 2022). In Appendix A.1, we briefly review the definition of value functions and the actor-critic framework in RL.

**Occupancy measures.** Alternatively, one can summarize the stochasticity over trajectories into the *discounted state occupancy measure* (Dayan, 1993; Eysenbach et al., 2022; Janner et al., 2020; Touati & Ollivier, 2021; Zheng et al., 2024b; Myers et al., 2024; Blier et al., 2021) that quantifies the discounted visitation frequency of different states under the policy $\pi$. Prior work (Dayan, 1993; Janner et al., 2020; Touati & Ollivier, 2021; Zheng et al., 2024b) has shown that the discounted state occupancy measure follows a Bellman equation backing up the probability density at the current time step and the future time steps:

$$p_\gamma^\pi(s_f \mid s, a) = (1 - \gamma)\delta_s(s_f) + \gamma \mathbb{E}_{\substack{s' \sim p(s'|s,a) \\ a' \sim \pi(a'|s')}} \left[ p_\gamma^\pi(s_f \mid s', a') \right], \tag{1}$$

where $\delta_s(\cdot)$ denotes the Dirac delta measure centered at $s$.[2] The discounted state occupancy measure allows us to rewrite the Q-function as a linear function of rewards (Barreto et al., 2017; Touati & Ollivier, 2021; Zheng et al., 2024b; Sikchi et al., 2024):

$$Q^\pi(s, a) = \frac{1}{1 - \gamma} \mathbb{E}_{s_f \sim p_\gamma^\pi(s_f|s,a)} \left[ r(s_f) \right]. \tag{2}$$

---

[2]The recursive relationship in Eq. 1 starts from the current time step (Eysenbach et al., 2022; Touati & Ollivier, 2021) instead of the next time step as in some prior approaches (Janner et al., 2020; Zheng et al., 2024b; Thakoor et al., 2022).

The alternative (dual (Sikchi et al., 2024)) definition of Q-function (Eq. 2) allows us to cast the policy evaluation step as first learning a generative model $p_\gamma(s_f \mid s, a)$ to simulate the discounted state occupancy measure of $\pi^k$ and then regressing the estimator $Q$ towards the average reward at states sampled from $p_\gamma$ (Toussaint & Storkey, 2006; Tomar et al., 2024; Thakoor et al., 2022; Zheng et al., 2024b). See Sec. 4.4 for detailed formulation.

**Flow matching and TD flows.** Flow matching (Lipman et al., 2023; 2024; Liu et al., 2023; Albergo & Vanden-Eijnden, 2023) refers to a family of generative models based on ordinary differential equations (ODEs), which are close cousins of denoising diffusion models (Sohl-Dickstein et al., 2015; Song et al., 2021; Ho et al., 2020), which instead solve a stochastic differential equation (SDE). The deterministic nature of ODEs equips flow-matching methods with more stable learning objectives and faster inference speed than denoising diffusion models (Lipman et al., 2023; 2024; Verine et al., 2023). In Appendix A.2, we discuss the problem setting and the standard learning objective for flow matching.

In the context of RL, prior work has used flow matching to estimate the discounted state occupancy measure (Farebrother et al., 2025) by incorporating the Bellman equation (Eq. 1) into the conditional flow matching loss (Eq. 10), resulting in a temporal difference flow matching procedure (TD flows) (Farebrother et al., 2025). In Appendix A.3, we discuss the detailed formulations of the TD flow objective for a target policy $\pi$. Choosing the target policy $\pi$ to be the same as the behavioral policy $\beta$, we obtain a SARSA (Rummery & Niranjan, 1994) variant of the loss optimizing the SARSA flows. We will use the SARSA variant of the TD flow objective to learn our generative occupancy models in Sec. 4.3.

# 4 INTENTION-CONDITIONED FLOW OCCUPANCY MODELS

In this section, we will introduce our method for pre-training and fine-tuning in RL. After formalizing the problem setting, we will dive into the latent variable model for pre-training an intention encoder and flow occupancy model. After pre-training the occupancy models, our method will extract polices for solving different tasks by invoking a generalized policy improvement procedure (Barreto et al., 2017). We refer to our method as **intention-conditioned flow occupancy models (InFOM)**.

## 4.1 PROBLEM SETTING

We consider learning with purely *offline* datasets, where an unlabeled (reward-free) dataset of transitions $D = \{(s, a, s', a')\}$ collected by the behavioral policy $\beta$ is provided for pre-training and a reward-labeled dataset $D_{\text{reward}} = \{(s, a, r)\}$ collected by some other policy $\tilde{\beta}$ on a downstream task is used for fine-tuning. Importantly, the behavioral policy $\beta$ used to collect $D$ can consist of a mixture of policies used by different users to complete distinct tasks. We will call this heterogeneous structure of the unlabeled datasets "intentions," which are latent vectors $z$s in some latent space $\mathcal{Z}$. In practice, these intentions can refer to desired goal images or language instructions that index the behavioral policy $\beta = \{\beta(\cdot \mid \cdot, z) : z \in \mathcal{Z}\}$. Because these latent intentions are *unobserved* to the pre-training algorithm, we want to infer them as a latent random variable $Z$ from the offline dataset, similar to prior work (Hausman et al., 2017; Li et al., 2017; Henderson et al., 2017). In Appendix B.2, we include discussions distinguishing our problem setting from meta RL and multi-task RL problems.

During pre-training, our method exploits the heterogeneous structure of the unlabeled dataset and extracts actionable information by *(1)* inferring intentions of the data collection policy and *(2)* learning occupancy models to predict long-horizon future states based on those intentions (Sec. 4.2 & 4.3). During fine-tuning, we first recover a set of intention-conditioned Q functions by regressing towards average rewards at future states generated by the occupancy models, and then extract a policy to maximize task-specific discounted cumulative returns (Sec. 4.4). Our method builds upon an assumption regarding the consistency of latent intentions.

**Assumption 1** (Consistency)**.** *The unlabeled dataset $D$ for pre-training is obtained by executing a behavioral policy following a mixture of unknown intentions $z \in \mathcal{Z}$. We assume that consecutive transitions $(s, a)$ and $(s', a')$ share the same intention.*

The consistency of intentions across transitions enables both intention inference using two sets of transitions and dynamic programming over trajectory segments. See Appendix B.1 for justifications of this assumption.

## 4.2 VARIATIONAL INTENTION INFERENCE

The goal of our pre-training framework is to learn a latent variable model that captures both long-horizon temporal information and unknown user intentions in the unlabeled datasets.

This part of our method aims to infer the intention $z$ based on consecutive transitions $(s, a, s', a')$ using the encoder $p_e(z \mid s', a')$ and predict the occupancy measures of a future state $s_f$ using the occupancy models $q_d(s_f \mid s, a, z)$. We want to maximize the likelihood of observing a future state $s_f$ starting from a state-action pair $(s, a)$ (amortized variational inference (Kingma & Welling, 2013; Margossian & Blei, 2024)), both sampled from the unlabeled dataset $D$ following the joint behavioral distribution $p^\beta(s, a, s_f) = p^\beta(s, a)p^\beta(s_f \mid s, a)$:

$$\max_{q_d} \mathbb{E}_{p^\beta(s,a,s_f)} \left[ \log q_d(s_f \mid s, a) \right]$$

$$\geq \max_{p_e, q_d} \mathbb{E}_{p^\beta(s,a,s_f,s',a')} \left[ \mathbb{E}_{p_e(z|s',a')} \left[ \log q_d(s_f \mid s, a, z) \right] - \lambda D_{\mathrm{KL}}(p_e(z \mid s', a') \parallel p(z)) \right], \quad (3)$$

where $p(z) = \mathcal{N}(0, I)$ denotes an uninformative standard Gaussian prior over intentions, $\lambda \geq 1$ denotes the coefficient that controls the strength of the KL divergence regularization. In practice, we can use any $\lambda \geq 0$ because rescaling the *input* $(s, a, s_f)$, similar to normalizing the range of images from $\{0, \cdots, 255\}$ to $[0, 1]$ in the original VAE (Kingma & Welling, 2013), preserves the ELBO. We defer the full derivation of the evidence lower bound (ELBO) and the explanation of $\lambda$ to Appendix C.1. Inferring the intention $z$ from the next transition $(s', a')$ follows from our consistency assumption (Assump. 1), and is important for avoiding overfitting (Frans et al., 2024). Importantly, $p_e$ and $q_d$ are optimized *jointly* with this objective. One way of understanding this ELBO is as maximizing an information bottleneck with the chain of random variables $(S', A') \to Z \to (S, A, S_f)$. See Appendix C.1 for the connection.

We use flow matching to reconstruct the discounted state occupancy measure rather than maximizing the likelihood directly, resulting in minimizing a surrogate objective:

$$\min_{p_e, q_d} \mathcal{L}_{\mathrm{Flow}}(q_d, p_e) + \lambda \mathbb{E}_{p^\beta(s',a')} \left[ D_{\mathrm{KL}}(p_e(z \mid s', a') \parallel p(z)) \right]. \quad (4)$$

We use $\mathcal{L}_{\mathrm{Flow}}$ to denote a placeholder for the flow matching loss and will instantiate this loss for the flow occupancy models $q_d$ next.

## 4.3 PREDICTING THE FUTURE VIA SARSA FLOWS

We now present the objective used to learn the flow occupancy models, where we first introduce some motivations and desiderata and then specify the actual loss. Given an unlabeled dataset $D$ and an intention encoder $p_e(z \mid s', a')$, the goal is to learn a *generative* occupancy model $q_d(s_f \mid s, a, z)$ that approximates the discounted state occupancy measure of the behavioral policy conditioned on different intentions, i.e., $q_d(s_f \mid s, a, z) \approx p^\beta(s_f \mid s, a, z)$. We will use $v_d : [0, 1] \times \mathcal{S} \times \mathcal{S} \times \mathcal{A} \times \mathcal{Z} \to \mathcal{S}$ to denote the time-dependent vector field that corresponds to $q_d$. There are two desired properties of the learned occupancy models: *(1)* distributing the peak probability density to multiple $s_f$, i.e., modeling multimodal structure, and *(2)* stitching together trajectory segments that share some transitions in the dataset, i.e., enabling combinatorial generalization. The first property motivates us to use an expressive flow-matching model (Lipman et al., 2024), while the second property motivates us to learn those occupancy models using temporal difference approaches (Janner et al., 2020; Tomar et al., 2024; Farebrother et al., 2025). Prior work (Farebrother et al., 2025) has derived the TD version of the regular (Monte Carlo) flow matching loss (Eq. 10) that incorporates the Bellman backup into the flow matching procedure, showing the superiority in sample efficiency and the capability of dynamic programming. We will adopt the same idea and use the SARSA variant of the TD flow loss

(Eq. 11) to learn our intention-conditioned flow occupancy models:

$$\mathcal{L}_{\text{SARSA flow}}(v_d, p_e) = (1 - \gamma)\mathcal{L}_{\text{SARSA current flow}}(v_d, p_e) + \gamma\mathcal{L}_{\text{SARSA future flow}}(v_d, p_e), \qquad (5)$$

$$\mathcal{L}_{\text{SARSA current flow}}(v_d, p_e) = \mathbb{E}_{\substack{(s,a,s',a')\sim p^\beta(s,a,s',a'), \\ z\sim p_e(z|s',a'), \\ t\sim\text{UNIF}([0,1]),\epsilon\sim\mathcal{N}(0,I)}} \left[ \|v(t, s^t, s, a, z) - (s - \epsilon)\|_2^2 \right],$$

$$\mathcal{L}_{\text{SARSA future flow}}(v_d, p_e) = \mathbb{E}_{\substack{(s,a,s',a')\sim p^\beta(s,a,s',a'), \\ z\sim p_e(z|s',a'), \\ t\sim\text{UNIF}([0,1]),\epsilon\sim\mathcal{N}(0,I)}} \left[ \|v_d(t, \bar{s}_f^t, s, a, z) - \bar{v}_d(t, \bar{s}_f^t, s', a', z)\|_2^2 \right].$$

Importantly, incorporating the information from latent intentions into the flow occupancy models allows us to *(1)* use the simpler and more stable SARSA bootstrap instead of the Q-learning style bootstrap (Eq. 11) on large datasets, *(2)* generalize over latent intentions, avoiding counterfactual errors. Sec. 5.2 visualizes the latent intentions, and Appendix F.2 contains additional experiments.

### 4.4    GENERATIVE VALUE ESTIMATION AND IMPLICIT GENERALIZED POLICY IMPROVEMENT

We next discuss the fine-tuning process in our algorithm. Our fine-tuning method builds on the dual perspective of value estimation introduced in the preliminaries (Eq. 2). We first estimate a *set* of intention-conditioned Q functions using regression and then use those intention-conditioned Q functions to extract a policy, utilizing generalized policy improvement (GPI) (Barreto et al., 2017). The key idea of GPI is that, in addition to taking the maximum over the actions, we can also take the maximum over the intentions. In our setting, the number of intentions is infinite—one for every choice of continuous $z$. Thus, taking the maximum over the intentions is both nontrivial and susceptible to instability (Sec. 5.3). We address this issue by replacing the greedy "max" with an upper expectile loss, resulting in an implicit generalized policy improvement procedure.

**Generative value estimation.** Given a reward-labeled dataset $D_{\text{reward}}$ and the pre-trained flow occupancy models $q_d$, we can estimate intention-conditioned Q values for a downstream task. Specifically, for a fixed latent intention $z \in \mathcal{Z}$, we first sample a set of $N$ future states from the flow occupancy models, $s_f^{(1)}, \cdots, s_f^{(N)} : s_f^{(i)} \sim q_d(s_f \mid s, a, z)$, and then constructs a Monte Carlo (MC) estimation of the Q function using those generative samples:[3]

$$Q_z(s, a) = \frac{1}{(1-\gamma)N} \sum_{i=1}^{N} r\left(s_f^{(i)}\right), \; s_f^{(i)} \sim q_d(s_f \mid s, a, z), \qquad (6)$$

where $r(\cdot)$ is the reward function or a learned reward predictor. Importantly, the choice of the number of future states $N$ affects the accuracy and variance of our Q estimate. Ablation experiments in Appendix F.11 indicate that $N = 16$ works effectively in our experiments. Note that we choose to sample $z$ from the prior $p(z)$ instead of from the posterior $q_d(z \mid s', a')$, resembling drawing random samples from a variational auto-encoder (Kingma & Welling, 2013). We include an ablation study in Appendix F.5, comparing the effect of fine-tuning with latents from the prior $p(z)$ and the posterior $q_d(z \mid s', a')$. In practice, we find sampling from the prior $p(z)$ worked well in our experiments.

**Implicit generalized policy improvement.** We can then use those MC estimation of Q functions to learn a policy by invoking the generalized policy improvement. The naive GPI requires sampling a finite set of latent intentions from the prior distribution $p(z)$, $z^{(1)}, \cdots, z^{(M)} : z^{(j)} \sim p(z)$ and greedily choose one $Q_z$ to update the policy (Barreto et al., 2017):

$$\arg\max_\pi \mathbb{E}_{\substack{s\sim p^{\tilde{\beta}}(s), \, a\sim\pi(a|s) \\ z^{(1)}, \cdots, z^{(M)}: z^{(j)}\sim p(z)}} \left[ \max_{z^{(j)}} Q_{z^{(j)}}(s, a) \right].$$

Despite its simplicity, the naive GPI suffers from two main disadvantages. First, using the maximum Q over a finite set of latent intentions to approximate the maximum Q over an infinite number of intentions results in local optima. Second, when we take gradients of this objective with respect to the policy, the chain rule gives one term involving $\nabla_a q_d(s_f|s, a, z)$. Thus, computing the gradients requires differentiating through the ODE solver (backpropagating through time (Park et al., 2025b)), which is unstable. We address these challenges by learning an explicit scalar Q function to distill the

---

[3]We omit the dependency of $Q_z$ on $s_f^{(1)}, \cdots, s_f^{(N)}$ to simplify notations.

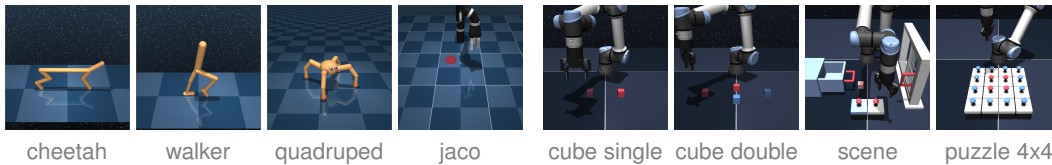

cheetah    walker    quadruped    jaco    cube single    cube double    scene    puzzle 4x4

Figure 2: **Domains for evaluation.** *(Left)* ExORL domains (16 state-based tasks). *(Right)* OGBench domains (20 state-based tasks and 4 image-based tasks).

MC estimation of intention-conditioned Q functions. This approach is appealing because gradients of the Q function no longer backpropagate through the ODE solver. We also replace the "max" over a finite set of intention-conditioned Q functions with an upper expectile loss $L_2^\mu$ (Kostrikov et al., 2022), resulting in the following critic loss

$$\mathcal{L}(Q) = \mathbb{E}_{(s,a)\sim p^{\tilde{\beta}}(s,a),\ z\sim p(z)} \left[ L_2^\mu \left( Q_z(s,a) - Q(s,a) \right) \right], \qquad (7)$$

where $L_2^\mu(x) = |\mu - \mathbb{1}(x < 0)| x^2$ and $\mu \in [0.5, 1)$. In Appendix C.2, we discuss the intuition and theoretical soundness of this distillation step. After distilling the intention-conditioned Q functions into a single function, we can extract the policy by selecting actions to maximize $Q$ with a behavioral cloning regularization (Fujimoto & Gu, 2021) using the actor loss

$$\mathcal{L}(\pi) = -\mathbb{E}_{(s,a)\sim p^{\tilde{\beta}}(s,a),a^\pi\sim\pi(a^\pi|s)}[Q(s,a^\pi) + \alpha \log \pi(a \mid s)], \qquad (8)$$

where $\alpha$ controls the regularization strength. We use the behavioral cloning regularization to both reduce errors from sampling out-of-distribution (OOD) actions (Kumar et al., 2020; Fujimoto & Gu, 2021) and mitigate error propagations through overestimated $Q_z$ values. Ablation experiments in Appendix F.7 and Appendix F.12 show that this behavioral cloning regularization is important for improving the policy performance. Taken together, we call the expectile Q distillation step (Eq. 7) and the policy optimization step (Eq. 8) *implicit generalized policy improvement (implicit GPI)*.

**Algorithm summary.** We use neural networks to parameterize the intention encoder $p_\phi$, the vector field of the occupancy models $v_\theta$, the reward predictor $r_\eta$, the critic $Q_\psi$, and the policy $\pi_\omega$. We consider two stages: pre-training and fine-tuning. In Alg. 1, we summarize the pre-training process of InFOM. InFOM pre-trains *(1)* the vector field $v_\theta$ using the SARSA flow loss (Eq. 5) and *(2)* the intention encoder $p_\phi$ using the ELBO (Eq. 3). Alg. 2 shows the pseudocode of InFOM for fine-tuning. InFOM mainly learns *(1)* the reward predictor $r_\eta$ via simple regression, *(2)* the critic $Q_\psi$ using expectile distillation (Eq. 7), and *(3)* the policy $\pi_\omega$ by conservatively maximizing the $Q_\psi$ (Eq. 8). The open-source implementation is available online.[4]

## 5 EXPERIMENTS

Our experiments start with comparing InFOM to prior methods that first pre-train on reward-free datasets and then fine-tune on reward-labeled datasets, measuring the performance on downstream tasks. We then study the two main components of our method: the variational intention encoder and the implicit GPI policy extraction strategy. Visualizations of the latent intention inferred by our variational intention encoder show alignment with the underlying ground-truth intentions. Our ablation experiments reveal the effect of the implicit GPI policy extraction strategy. We also include additional experiments showing InFOM enables faster policy learning during fine-tuning in Appendix F.3. Our algorithm is robust to various choices of hyperparameters (Appendix F.12). Following prior work (Park et al., 2025b), all experiments report means and standard deviations across 8 random seeds for state-based tasks and 4 random seeds for image-based tasks.

### 5.1 COMPARING TO PRIOR PRE-TRAINING AND FINE-TUNING METHODS

Our experiments study whether the proposed method (InFOM), which captures actionable information conditioned on user intentions from unlabeled datasets, enables effective pre-training and fine-tuning. We select 36 state-based and 4 image-based tasks across diverse robotic navigation and manipulation

---

[4]https://github.com/chongyi-zheng/infom

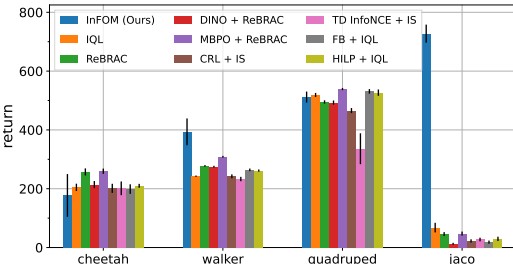 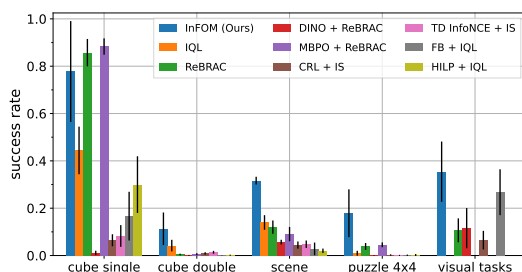

(a) 16 state-based ExORL tasks from Yarats et al. (2022). We average over 4 tasks for each domain.

(b) 20 state-based and 4 image-based OGBench tasks from Park et al. (2025a). We average over 5 tasks for each state-based domain and average over 4 visual tasks.

Figure 3: **Evaluation on ExORL and OGBench tasks.** We compare InFOM against prior methods that utilize various learning paradigms on task-agnostic pre-training and task-specific fine-tuning. InFOM performs similarly to, if not better than, prior methods on 7 out of the 9 domains, including the most challenging visual tasks. We report means and standard deviations over 8 random seeds (4 random seeds for image-based tasks) with error bars indicating one standard deviation. See Table 4 for full results.

domains and compare against 8 baselines (Fig. 2). The models pre-trained by those methods include behavioral cloning policies (Kostrikov et al., 2022; Tarasov et al., 2023a), transition models (Janner et al., 2019), representations (Caron et al., 2021), discriminative classifiers that predict occupancy measures (Eysenbach et al., 2022; Zheng et al., 2024b), and latent skills (Touati & Ollivier, 2021; Park et al., 2024b). We defer the detailed discussions about benchmarks and datasets to Appendix D.1 and the rationale for choosing different baselines to Appendix D.2. Whenever possible, we use the same hyperparameters for all methods (Table 1). See Appendix D.3 for details of the evaluation protocol and Appendix D.4 for implementations and hyperparameters of each method.

We report results in Fig. 3, aggregating over four tasks in each domain of ExORL and five tasks in each domain of OGBench, and present the full results in Table 4. These results show that InFOM matches or surpasses all baselines on six out of eight domains. On ExORL benchmarks, all methods perform similarly on the two easier domains (cheetah and quadruped), while InFOM can obtain $20\times$ improvement on jaco, where baselines only make trivial progress (Table 4). We suspect the outsized improvement on the jaco task is because of the high-dimensional state space (twice that of the other ExORL tasks (Yarats et al., 2022)) and because it has sparse rewards; Appendix Fig. 13 supports this hypothesis by showing that the ReBRAC baseline achieves significantly higher returns when using dense rewards. On those more challenging state-based manipulation tasks from OGBench, we find a marked difference between baselines and InFOM; our method achieves $36\%$ higher success rate over the best performing baseline. In addition, InFOM is able to outperform the best baseline by $31\%$ using RGB images as input directly (visual tasks). We hypothesize that the baselines fail to solve these more challenging tasks because of the semi-sparse reward functions. In contrast, our method can explore different regions of the state space using the different intentions, thereby addressing the challenge of reward sparsity. We conjecture that the variance of InFOM across seeds in some experiments (e.g., cheetah, cube single, and puzzle 4x4) reflects stochasticity in the MC Q estimates (Eq. 6), which might be mitigated by increasing the number of sampled future states (See Appendix F.11). In Appendix F.1, we compare InFOM against selective baselines on real robotics datasets, showing $34\%$ improvement.

## 5.2 VISUALIZING LATENT INTENTIONS

Our next experiment studies the intention encoder in our algorithm. To investigate whether the proposed method discovers distinct user intentions from an unlabeled dataset, we visualize latent intentions inferred by our variational intention encoder. We include comparisons against two alternative intention encoding mechanisms proposed by prior methods. Specifically, we consider replacing the variational intention encoder with either *(1)* a set of Hilbert representations (Park et al., 2024b) or *(2)* a set of forward-backward representations (Touati & Ollivier, 2021), and then pre-training the flow occupancy models (FOM) conditioned on these two sets of representations. We call these two variants HILP + FOM and FB + FOM. Note that FB + FOM is equivalent to TD flows with GPI in Farebrother et al. (2025). Using t-SNE (Maaten & Hinton, 2008), we visualize latent intentions predicted by these three methods on cube double task 1 from the OGBench benchmarks.

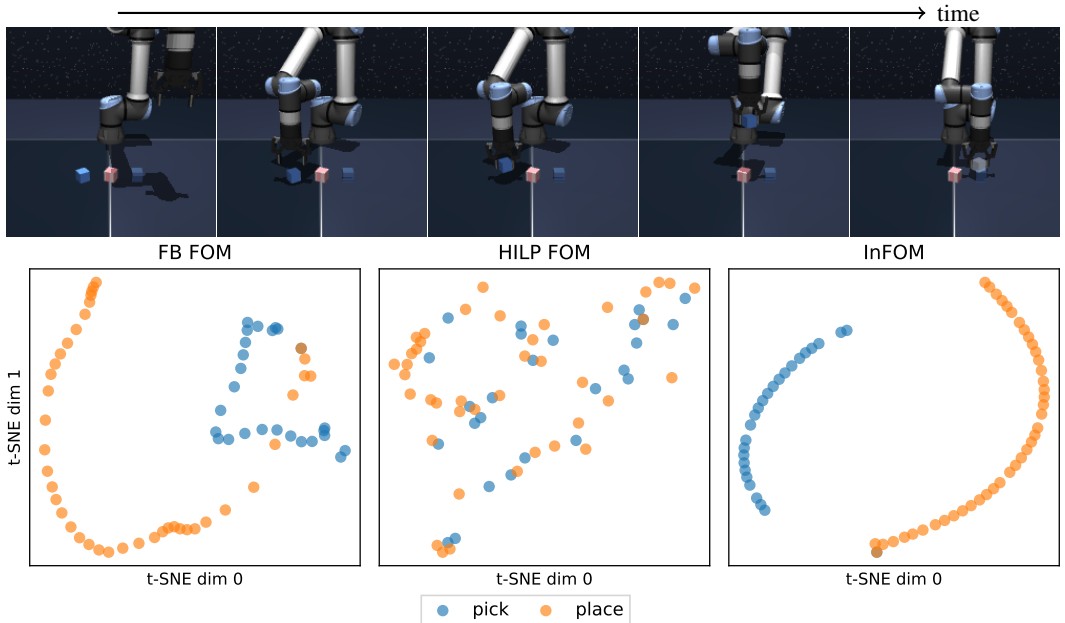

Figure 4: **Visualization of latent intentions.** *(Top)* The optimal policy picks up the blue block from the left and places it on the right. *(Bottom)* Using t-SNE (Maaten & Hinton, 2008), we visualize the latent intentions inferred by the variational intention encoder in InFOM, comparing against latent representations inferred by HILP and FB for learning FOMs. The predictions from InFOM align with the underlying intentions. See Sec. 5.2 for details and Appendix E for more visualizations.

Fig. 4 shows the optimal trajectory, where the manipulator picks the blue block from the left and then places it on the right, and the visualizations. The 2D t-SNE visualizations indicate that both FB + FOM and HILP + FOM infer mixed latent intentions for "pick" and "place" behaviors, while InFOM predicts a sequence of latent intentions with clear clustering. This result suggests that InFOM is capable of inferring latent intentions that align with the underlying ground-truth intentions. See Appendix E for more visualizations. In Appendix F.2, we include additional experiments comparing the downstream performance between InFOM and HILP + FOM and FB + FOM. Results in Appendix Fig. 9 suggest that InFOM can outperform those two baselines on 3 of 4 tasks.

### 5.3 IMPORTANCE OF THE IMPLICIT GENERALIZED POLICY IMPROVEMENT

Our final experiments study different approaches for policy optimization. We hypothesize that our proposed method is more efficient and robust than other policy extraction strategies. To test this hypothesis, we conduct ablation experiments on one task in the ExORL benchmarks (`quadruped jump`) and another task taken from the OGBench benchmarks (`scene task 1`), again following the evaluation protocols in Appendix D.3. We compare two alternative policy learning approaches in the fine-tuning phase. First, we ablate the effect of the upper expectile loss by comparing against the standard GPI, which maximizes Q functions over a finite set of intentions $\{z^{(1)}, \cdots, z^{(M)}\}$. We choose

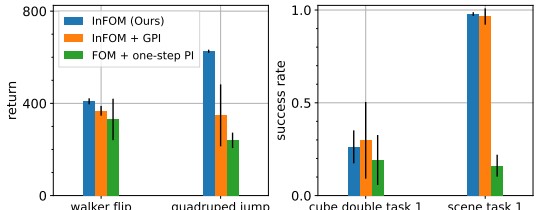

Figure 5: **Comparison to alternative policy extraction strategies.** We compare InFOM to alternative policy extraction strategies based on the standard generalized policy improvement or one-step policy improvement. Our method is 44% more performant with 8× smaller variance than the variant using the standard GPI. See Sec. 5.3 for details.

$M = 32$ latent intentions to balance between performance and compute budget, and call this variant InFOM + GPI. Second, we ablate the effect of the variational intention encoder by removing the intention dependency in the flow occupancy models and extracting the policy via one-step policy improvement (PI) (Wang et al., 2018; Brandfonbrener et al., 2021; Peters & Schaal, 2007; Peters et al., 2010). We call this method FOM + one-step PI and defer the detailed formulation to Appendix C.3.

As shown in Fig. 5, InFOM achieves significantly higher returns and success rates than its variant based on one-step policy improvement, suggesting the importance of inferring user intentions.

Compared with its GPI counterpart, our method is $44\%$ more performant with $8\times$ smaller variance (the error bar indicates one standard deviation), demonstrating that the implicit GPI indeed performs a relaxed maximization over intentions while maintaining robustness.

**Additional experiments.** In Appendix F.3, we include additional ablations showing that InFOM enables faster policy learning. Appendix F.4 ablates InFOM against a variant of InFOM with a set of discrete latents trained vector quantization loss, showing that the continous latent space generally performs better. In Appendix F.8, we relate the diversity of the pre-training datasets to their sizes. The dataset size ablations in Appendix F.9 show that using sufficient pre-training and fine-tuning data is important. Appendix F.10 studies the effects of fine-tuning on suboptimal datasets. Our hyperparameter ablations can be found in Appendix F.12.

**Alternative generative occupancy models.** Farebrother et al. (2025) has already discussed using alternative prior generative modeling approaches to learn the occupancy measure. Specifically, they compare flow-based occupancy models against representative generative methods, including denoising diffusion (Ho et al., 2020), VAE (Kingma & Welling, 2013; Higgins et al., 2017), and GAN (Goodfellow et al., 2014). Results in Fig. 2 of Farebrother et al. (2025) show that flow-based occupancy models ($TD^2$-CFM in the figure) outperform alternative generative methods in modeling the occupancy measures. For this reason, we do not include comparisons against alternative generative occupancy models to distinguish our contributions.

## 6 CONCLUSION

In this work, we presented InFOM, a method that captures diverse intentions and their long-term behaviors from an unstructured dataset, leveraging the expressivity of flow models. We empirically showed that the intentions captured in flow occupancy models enable effective and efficient fine-tuning, outperforming prior unsupervised pre-training approaches on diverse state- and image-based domains.

**Limitations.** One limitation of InFOM is that our reduction from trajectories to consecutive state-action pairs might not always accurately capture the original intentions in the trajectories. While we empirically showed that this simple approach is sufficient to achieve strong performance on our benchmark tasks, it can be further improved with alternative trajectory encoding techniques and data collection strategies, which we leave for future work.

## REPRODUCIBILITY STATEMENT

We implement InFOM and all baselines in the same codebase using JAX (Bradbury et al., 2018). Our implementations build on top of OGBench's and FQL's implementations (Park et al., 2025a;b). We include the common hyperparameters for all the methods in Appendix Table 1, the hyperparameters for InFOM in Appendix Table 2 and Appendix Table 3, and the hyperparameters for baselines in Appendix Table 3. All the experiments were run on a single NVIDIA A6000 GPU and can be finished in 4 hours for state-based tasks and 12 hours for image-based tasks. We provide open-source implementations of InFOM and all baselines at https://github.com/chongyi-zheng/infom.

## ACKNOWLEDGMENTS

We thank Kevin Frans, Homer Walker, Aravind Venugopal, and Bogdan Mazoure for their helpful discussions. We also thank Tongzhou Wang for sharing code during the early stages of our experiments. This work used the Della computational cluster provided by Princeton Research Computing, as well as the Ionic and Neuronic computing clusters maintained by the Department of Computer Science at Princeton University. This research was supported by ONR N00014-22-1-2773 and was partly supported by the Korea Foundation for Advanced Studies (KFAS). This material is based upon work supported by the National Science Foundation under Award No. 2441665. Any opinions, findings, conclusions, or recommendations expressed in this material are those of the authors and do not necessarily reflect the views of the National Science Foundation.

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

---

**Algorithm 1** Intention-Conditioned Flow Occupancy Model (pre-training).

---

1: **Input** The intention encoder $p_\phi$, the vector field $v_\theta$, the target vector field $v_{\bar\theta}$, the policy $\pi_\omega$, and the reward-free dataset $D$.
2: **for** each iteration **do**
3:      Sample a batch of $\{(s, a, s', a') \sim D\}$.
4:      Sample a batch of $\{\epsilon \sim \mathcal{N}(0, I)\}$ and a batch of $\{t \sim \text{UNIF}([0, 1])\}$.
5:      Encode intentions $\{z \sim p_\phi(z \mid s', a')\}$ for each $(s', a')$.
     ▽ SARSA flow occupancy model loss.
6:      $s^t \leftarrow (1 - t)\epsilon + ts$
7:      $\bar{s}_f \leftarrow \text{EulerMethod}(v_{\bar\theta}, \epsilon, s', a', z)$, $\bar{s}_f^t \leftarrow (1 - t)\epsilon + t\bar{s}_f$.
8:      $\mathcal{L}_{\text{SARSA current flow}}(\theta, \phi) \leftarrow \mathbb{E}_{(s,a,z,t,\epsilon,s^t)} \left[ \|v_\theta(t, s^t, s, a, z) - (s - \epsilon)\|_2^2 \right]$.
9:      $\mathcal{L}_{\text{SARSA future flow}}(\theta, \phi) \leftarrow \mathbb{E}_{(s,a,z,t,\epsilon,\bar{s}_f^t)} \left[ \|v_\theta(t, \bar{s}_f^t, s, a, z) - v_{\bar\theta}(t, \bar{s}_f^t, s', a', z)\|_2^2 \right]$.
10:      $\mathcal{L}_{\text{SARSA flow}}(\theta, \phi) \leftarrow (1 - \gamma)\mathcal{L}_{\text{current}}(\theta, \phi) + \gamma\mathcal{L}_{\text{future}}(\theta, \phi)$.      ▷ Eq. 5
     ▽ Intention encoder loss.
11:      $\mathcal{L}_{\text{ELBO}}(\theta, \phi) \leftarrow \mathcal{L}_{\text{SARSA flow}}(\theta, \phi) + \lambda\mathbb{E}_{(s',a')} \left[ D_{\text{KL}} \left( p_\phi(z \mid s', a') \,\|\, \mathcal{N}(0, I) \right) \right]$.      ▷ Eq. 4
     ▽ (Optional) Behavioral cloning loss.
12:      $\mathcal{L}_{\text{BC}}(\omega) \leftarrow -\mathbb{E}_{(s,a)} \left[ \log \pi_\omega(a \mid s) \right]$.
13:      Update the vector field $\theta$ and the intention encoder $\phi$ by minimizing $\mathcal{L}_{\text{ELBO}}(\theta, \phi)$.
14:      Update the policy $\omega$ by minimizing $\mathcal{L}_{\text{BC}}(\omega)$.
15:      Update the target vector field $\bar\theta$ using the Polyak average of $\theta$.
16: **Return** $v_\theta$, $p_\phi$, and $\pi_\omega$.

---

# A   PRELIMINARIES

## A.1   VALUE FUNCTIONS AND THE ACTOR-CRITIC FRAMEWORK

The goal of RL is to learn a policy $\pi : \mathcal{S} \to \Delta(\mathcal{A})$ that maximizes the expected discounted return $J(\pi) = \mathbb{E}_{\tau \sim \pi(\tau)} \left[ \sum_{h=0}^{\infty} \gamma^h r_h \right]$, where $\tau$ is a trajectory sampled by the policy. We will use $\beta : \mathcal{S} \to \Delta(\mathcal{A})$ to denote the behavioral policy. Given a policy $\pi$, we measure the expected discounted return starting from a state-action pair $(s, a)$ and a state $s$ as the (unnormalized) Q-function and the value function, respectively:

---

**Algorithm 3** Euler method for solving the flow ODE (Eq. 9).

---

1: **Input** The vector field $v$ and the noise $\epsilon$. (Optional) The number of steps $T$ with default $T = 10$.
2: Initialize $t = 0$ and $x^t = \epsilon$
3: **for** each step $t = 0, 1, \cdots, T - 1$ **do**
4:      $x^{t+1} \leftarrow x^t + v(t/T, x^t)/T$
5: **Return** $\hat{x} = x^T$

---

$$Q^\pi(s, a) = \mathbb{E}_{\tau \sim \pi(\tau)} \left[ \sum_{h=0}^{\infty} \gamma^h r_h \,\middle|\, s_0 = s, a_0 = a \right], \quad V^\pi(s) = \mathbb{E}_{a \sim \pi(a|s)} \left[ Q^\pi(s, a) \right].$$

Prior actor-critic methods (Schulman et al., 2015; 2017; Haarnoja et al., 2018; Fujimoto et al., 2018; Kumar et al., 2020; Fujimoto & Gu, 2021) typically maximize the RL objective $J(\pi)$ by *(1)* learning an estimate $Q$ of $Q^\pi$ via the temporal difference (TD) loss (policy evaluation) and then *(2)* improving the policy $\pi$ by selecting actions that maximizes $Q$ (policy improvement):

$$Q^{k+1} \leftarrow \arg\max_Q \mathbb{E}_{(s,a,r,s') \sim p^\beta(s,a,r,s'), a' \sim \pi^k(a'|s')} \left[ \left( Q(s, a) - (r + \gamma Q^k(s', a')) \right)^2 \right]$$

$$\pi^{k+1} \leftarrow \arg\max_\pi \mathbb{E}_{s \sim p^\beta(s), a \sim \pi(a|s)} \left[ Q^{k+1}(s, a) \right],$$

where $k$ indicates the number of updates and $\beta$ is the behavioral policy representing either a replay buffer (online RL) or a fixed dataset (offline RL).

---

**Algorithm 2** Intention-Conditioned Flow Occupancy Model (fine-tuning).

---

1: **Input** The intention encoder $p_\phi$, the vector field $v_\theta$, the target vector field $v_{\bar{\theta}}$, the reward predictor $r_\eta$, the critic $Q_\psi$, the policy $\pi_\omega$ (random initialization or initialized using $\pi_\omega$ from Alg. 1), and the reward-labeled dataset $D_{\text{reward}}$.
2: **for** each iteration **do**
3:     Sample a batch of $\{(s, a, r, s', a') \sim D_{\text{reward}}\}$.
4:     Sample a batch of $\{\epsilon \sim \mathcal{N}(0, I)\}$ and a batch of $\{t \sim \text{UNIF}([0, 1])\}$.
5:     Sample prior intentions $\{z \sim p(z)\}$.
6:     Sample a batch of $\{(\epsilon^{(1)}, \cdots, \epsilon^{(N)}) \sim (\mathcal{N}(0, I), \cdots, \mathcal{N}(0, I))\}$.
        ▽ SARSA flow occupancy model loss and intention encoder loss.
7:     $\mathcal{L}_{\text{ELBO}}(\theta, \psi)$ as in Alg. 1.
        ▽ Reward predictor loss.
8:     $\mathcal{L}_{\text{Reward}}(\eta) \leftarrow \mathbb{E}_{(s,r)}\left[(r_\eta(s) - r)^2\right]$.
        ▽ Critic loss.
9:     $s_f^{(i)} \leftarrow \text{EulerMethod}(v_\theta, \epsilon^{(i)}, s, a, z)$ (Alg. 3) for each $(s, a, z, \epsilon^{(i)})$.
10:    $Q_z(s, a) \leftarrow \frac{1}{(1-\gamma)N} \sum_{i=1}^{N} r_\eta\left(s_f^{(i)}\right)$.                              ▷ Eq. 6
11:    $\mathcal{L}_{\text{Critic}}(\psi) \leftarrow \mathbb{E}_{\left(s,a,z,s_f^{(1)},\cdots,s_f^{(N)}\right)}\left[L_2^\mu\left(Q_z(s, a) - Q_\psi(s, a)\right)\right]$.                              ▷ Eq. 7
        ▽ Actor loss.
12:    $\mathcal{L}_{\text{Actor}}(\omega) \leftarrow -\mathbb{E}_{(s,a), a^\pi \sim \pi_\omega(a^\pi | s)}\left[Q_\psi(s, a^\pi) + \alpha \log \pi_\omega(a \mid s)\right]$.                              ▷ Eq. 8
13:    Update the vector field $\theta$ and the intention encoder $\phi$ by minimizing $\mathcal{L}_{\text{ELBO}}(\theta, \phi)$.
14:    Update the reward predictor $\eta$, the critic $\psi$, and the policy $\omega$ by minimizing $\mathcal{L}_{\text{Reward}}(\eta)$, $\mathcal{L}_{\text{Critic}}(\psi)$, and $\mathcal{L}_{\text{Actor}}(\omega)$ respectively.
15:    Update the target vector field $\bar{\theta}$ using the Polyak average of $\theta$.
16: **Return** $v_\theta$, $p_\phi$, $r_\eta$, $Q_\phi$, and $\pi_\omega$.

---

## A.2 FLOW MATCHING

The goal of flow matching methods is to transform a simple noise distribution (e.g., a $d$-dimensional standard Gaussian) into a target distribution $p_\mathcal{X}$ over some space $\mathcal{X} \subset \mathbb{R}^d$ that we want to approximate. Specifically, flow matching uses a time-dependent vector field $v : [0, 1] \times \mathbb{R}^d \to \mathbb{R}^d$ to construct a time-dependent diffeomorphic flow $\phi : [0, 1] \times \mathbb{R}^d \to \mathbb{R}^d$ (Lipman et al., 2023; 2024) that realizes the transformation from a single noise $\epsilon$ to a generative sample $\hat{x}$, following the ODE

$$\frac{d}{dt}\phi(t, \epsilon) = v(t, \phi(t, \epsilon)), \ \phi(0, \epsilon) = \epsilon, \ \phi(1, \epsilon) = \hat{x}. \tag{9}$$

We will use $t$ to denote a time step for flow matching and sample the noise $\epsilon$ from a standard Gaussian distribution $\mathcal{N}(0, I)$ throughout our discussions.[5] Prior work has proposed various formulations for learning the vector field (Lipman et al., 2023; Campbell et al., 2024; Liu et al., 2023; Albergo & Vanden-Eijnden, 2023) and we adopt the simplest flow matching objective building upon optimal transport (Liu et al., 2023) and conditional flow matching (CFM) (Lipman et al., 2023),

$$\mathcal{L}_{\text{CFM}}(v) = \mathbb{E}_{\substack{t \sim \text{UNIF}([0,1]), \\ x \sim p_\mathcal{X}(x), \epsilon \sim \mathcal{N}(0,I)}}\left[\|v(t, x^t) - (x - \epsilon)\|_2^2\right], \tag{10}$$

where $\text{UNIF}([0, 1])$ is the uniform distribution over the unit interval and $x^t = tx + (1 - t)\epsilon$ is a linear interpolation between the ground-truth sample $x$ and the Gaussian noise $\epsilon$. Importantly, we can generate a sample from the vector field $v$ by numerically solving the ODE (Eq. 9). We will use the Euler method (Alg. 3) as our ODE solver following prior practice (Grathwohl et al., 2019; Chen et al., 2018; Lipman et al., 2023; Liu et al., 2023; Park et al., 2025b; Frans et al., 2025).

---

[5] In theory, the noise can be drawn from any distribution, not necessarily limited to a Gaussian (Liu et al., 2023).

## A.3 TEMPORAL DIFFERENCE FLOWS

Given a policy $\pi$, prior work (Farebrother et al., 2025) models the occupancy measure $p_\gamma^\pi$ by optimizing the vector field $v : [0,1] \times \mathcal{S} \times \mathcal{S} \times \mathcal{A} \to \mathcal{S}$ using the following loss:

$$\mathcal{L}_{\text{TD flow}}(v) = (1-\gamma)\mathcal{L}_{\text{TD current flow}}(v) + \gamma\mathcal{L}_{\text{TD future flow}}(v) \tag{11}$$

$$\mathcal{L}_{\text{TD current flow}}(v) = \mathbb{E}_{\substack{t\sim\text{UNIF}([0,1]),\epsilon\sim\mathcal{N}(0,I),\\(s,a)\sim p^\beta(s,a)}} \left[\|v(t, s^t, s, a) - (s-\epsilon)\|_2^2\right]$$

$$\mathcal{L}_{\text{TD future flow}}(v) = \mathbb{E}_{\substack{t\sim\text{UNIF}([0,1]),\epsilon\sim\mathcal{N}(0,I),\\(s,a,s')\sim p^\beta(s,a,s'),a'\sim\pi(a'|s')}} \left[\|v(t, \bar{s}_f^t, s, a) - \bar{v}(t, \bar{s}_f^t, s', a')\|_2^2\right],$$

where $p^\beta(s,a)$ and $p^\beta(s,a,s')$ denote the joint distribution of transitions, $s^t = ts + (1-t)\epsilon$ is a linear interpolation between the current state $s$ and the noise $\epsilon$, and $\bar{v}$ denotes the Polyak average of historical $v$ over iterations (a target vector field) (Grill et al., 2020; Mnih et al., 2015; Caron et al., 2021). Of particular note is that we obtain a target future state $\bar{s}_f$ by applying the Euler method (Alg. 3) to $\bar{v}$ at the next state-action pair $(s', a')$, where $a'$ is sampled from the target policy $\pi$ of interest, and the noisy future state $\bar{s}_f^t = t\bar{s}_f + (1-t)\epsilon$ is a linear interpolation between this future state $\bar{s}_f$ and the noise $\epsilon$. Intuitively, minimizing $\mathcal{L}_{\text{TD current flow}}$ reconstructs the distribution of current state $s$, while minimizing $\mathcal{L}_{\text{TD future flow}}$ bootstraps the vector field $v$ at a noisy target future state $\bar{s}_f^t$, similar to Q-learning (Watkins & Dayan, 1992). Choosing the target policy $\pi$ to be the same as the behavioral policy $\beta$, we obtain a SARSA (Rummery & Niranjan, 1994) variant of the loss optimizing the SARSA flows. We call the loss in Eq. 11 the TD flow loss[6] and use the SARSA variant of it to learn generative occupancy models.

## B FURTHER DISCUSSIONS ON THE PROBLEM SETTING

### B.1 THE CONSISTENCY ASSUMPTION ON INTENTIONS

We now discuss the reason for making the consistency assumption (Assumption 1) on latent intentions. Since we use a heterogeneous behavioral policy to collect the unlabeled dataset, each unknown user intention indexed their own behavioral policy $\beta : \mathcal{S} \times \mathcal{Z} \to \Delta(\mathcal{A})$. The key observation is that the occupancy measure of each intention-conditioned behavioral policy follows its own Bellman equations (Similar to Eq. 1):

$$p_\gamma^\beta(s_f \mid s, a, z) = (1-\gamma)\delta_s(s_f) + \gamma\mathbb{E}_{\substack{s'\sim p(s'|s,a),\\a'\sim\beta(a'|s',z)}} \left[p_\gamma^\beta(s_f \mid s', a', z)\right],$$

suggesting that the same latent $z$ propagates through the transitions with the same underlying user intentions. Importantly, this propagation requires using a TD loss to estimate the behavioral occupancy measure, which aligns with the goal of our SARSA flow-matching losses (Eq. 5). We note that prior work (Touati & Ollivier, 2021) also adapts the same formulation of the intention-conditioned occupancy measure for zero-shot RL.

### B.2 DISTINCTIONS FROM META RL AND MULTI-TASK RL

Our problem setting is conceptually similar to meta RL (Duan et al., 2016; Rakelly et al., 2019; Pong et al., 2022) with two key distinctions. First, offline meta RL methods typically have access to explicit task descriptions (e.g., a one-hot task indicator) together with task-specific datasets. These descriptions and datasets induce a clear clustering of transitions. In contrast, our method must infer this structure from a heterogeneous dataset in an unsupervised manner. Second, offline meta RL trains on reward-labeled data during the meta-training phase, where task-specific rewards provide supervision for policy learning. In contrast, during pre-training, our method learns a generative model that predicts future states from inferred intentions without using any task-specific reward signals.

Similar to the distinctions between our setting and offline meta RL problems, our method does *not* fall into the multi-task RL category (Sodhani et al., 2021; Yu et al., 2020). During pre-training, *(1)* InFOM does not have access to task descriptions or task-specific datasets, and *(2)* it does not use any supervision from task-specific reward signals. Instead, InFOM pre-trains a generative, multi-step transition model that facilitates value estimation for downstream tasks.

---

[6]The TD flow loss is called the TD$^2$-CFM loss in Farebrother et al. (2025), and we rename it for simplicity.

## C  Theoretical analyses

### C.1  The evidence lower bound and its connection with an information bottleneck

We first derive the evidence lower bound for optimizing the latent variable model and then show its connection with an information bottleneck. Given the unlabeled dataset $D$, we want to maximize the likelihood of consecutive transitions $(s, a, s', a')$ and a future state $s_f$ sampled from the same trajectory following the behavioral joint distribution $p^\beta(s, a, s_f, s', a') = p^\beta(s)\beta(a \mid s)p_\gamma^\beta(s_f \mid s, a)p(s' \mid s, a)\beta(a' \mid s')$. We use $(s', a')$ to encode the intention $z$ by the encoder $p_e(z \mid s, a)$ and $(s, a, s_f, z)$ to learn the occupancy models $q_d(s_f \mid s, a, z)$, employing an ELBO of the likelihood of the prior data:

$$\mathbb{E}_{p^\beta(s,a,s_f)}\left[\log q_d(s_f \mid s, a)\right]$$

$$= \mathbb{E}_{p^\beta(s,a,s_f,s',a')}\left[\log q_d(s_f \mid s, a)\right]$$

$$= \mathbb{E}_{p^\beta(s,a,s_f,s',a')}\left[\log \mathbb{E}_{p(z)}\left[q_d(s_f \mid s, a, z)\right]\right]$$

$$\stackrel{(a)}{=} \mathbb{E}_{p^\beta(s,a,s_f,s',a')}\left[\log \mathbb{E}_{p(z)}\left[q_d(s_f \mid s, a, z)\frac{p_e(z \mid s', a')}{p_e(z \mid s', a')}\right]\right]$$

$$\stackrel{(b)}{\geq} \mathbb{E}_{p^\beta(s,a,s_f,s',a')}[\mathbb{E}_{p_e(z\mid s',a')}[\log q_d(s_f \mid s, a, z)] - D_{\mathrm{KL}}(p_e(z \mid s', a') \parallel p(z))]$$

$$\stackrel{(c)}{\geq} \mathbb{E}_{p^\beta(s,a,s_f,s',a')}[\mathbb{E}_{p_e(z\mid s',a')}[\log q_d(s_f \mid s, a, z)] - \lambda D_{\mathrm{KL}}(p_e(z \mid s', a') \parallel p(z))]$$

$$= \mathrm{ELBO}(p_e, q_d),$$

where in *(a)* we introduce the amortized variational encoder $p_e(z \mid s', a')$, in *(b)* we apply the Jensen's inequality (Durrett, 2019), and in *(c)* we introduce a coefficient $\lambda \geq 1$ to control the strength of the KL divergence regularization. In practice, we can use any $\lambda \geq 0$ because rescaling the *input* $(s, a, s_f)$, similar to normalizing the range of images from $\{0, \cdots, 255\}$ to $[0, 1]$ in the original VAE (Kingma & Welling, 2013), preserves the ELBO. Formally, following prior work (Higgins et al., 2017), maximizing this ELBO can also be interpreted as an optimization problem that simultaneously predicts future states while penalizing the intention encoder:

$$\max_{p_e,q_d} \mathbb{E}_{p^\beta(s,a,s_f,s',a')}[\mathbb{E}_{p_e(z\mid s',a')}[\log q_d(s_f \mid s, a, z)] \quad \text{s.t. } D_{\mathrm{KL}}(p_e(z \mid s', a') \parallel p(z)) \leq \text{const.}.$$

Rewriting this constrained optimization problem as the Lagrangian produces

$$\mathbb{E}_{p^\beta(s,a,s_f,s',a')}[\mathbb{E}_{p_e(z\mid s',a')}[\log q_d(s_f \mid s, a, z)] - \lambda D_{\mathrm{KL}}(p_e(z \mid s', a') \parallel p(z)),$$

where we introduce a coefficient $\lambda \geq 0$ to control the strength of the KL divergence regularization.

Alternatively, the constrained optimization problem can also be cast as a variational lower bound on an information bottleneck with the chain of random variables $(S', A') \to Z \to (S, A, S_f)$ (Tishby et al., 2000; Alemi et al., 2017; Saxe et al., 2018):

$$I^\beta(S, A, S_f; Z) - \lambda I^\beta(S', A'; Z)$$

$$\stackrel{(a)}{=} I^\beta(S, A, S_f; Z) - \lambda \mathbb{E}_{p^\beta(s',a')}\left[D_{\mathrm{KL}}(p_e(z \mid s', a') \parallel p_e(z))\right]$$

$$\stackrel{(b)}{\geq} I^\beta(S, A, S_f; Z) - \lambda \mathbb{E}_{p^\beta(s',a')}\left[D_{\mathrm{KL}}(p_e(z \mid s', a') \parallel p(z))\right]$$

$$\stackrel{(c)}{\geq} \mathbb{E}_{\substack{p^\beta(s,a,s_f,s',a') \\ p_e(z\mid s',a')}}\left[\log q_d(s, a, s_f \mid z)\right] - \lambda \mathbb{E}_{p^\beta(s',a')}\left[D_{\mathrm{KL}}(p_e(z \mid s', a') \parallel p(z))\right] + H^\beta(S, A, S_f)$$

$$\stackrel{(d)}{\geq} \mathbb{E}_{\substack{p^\beta(s,a,s_f,s',a') \\ p_e(z\mid s',a')}}\left[\log q_d(s_f \mid s, a, z)\right] - \lambda \mathbb{E}_{p^\beta(s',a')}\left[D_{\mathrm{KL}}(p_e(z \mid s', a') \parallel p(z))\right] + \text{const.}$$

where in *(a)* we use the definition of $I^\beta(S', A'; Z)$ and $p_e(z)$ is the marginal distribution of latent intentions $z$ defined as $p_e(z) = \int p^\beta(s', a')p_e(z \mid s', a')ds'da'$, in *(b)* we apply the non-negative property of the KL divergence $D_{\mathrm{KL}}(p_e(z) \parallel p(z))$, in *(c)* we apply the standard variation lower bound of the mutual information (Barber & Agakov, 2004; Poole et al., 2019) to incorporate the

decoder (occupancy models) $q_d(s, a, s_f \mid z)$, and in *(d)* we choose the variational decoder to satisfy $\log q_d(s, a, s_f \mid z) = \log p^\beta(s, a) + \log q_d(s_f \mid s, a, z)$ and consider the entropy $H^\beta(S, A, S_f)$ as a constant. Therefore, the lower bound in Eq. 3 can also be interpreted as maximizing the information bottleneck $I^\beta(S, A, S_f; Z) - \lambda I^\beta(S', A'; Z)$ with $\lambda \geq 0$.

### C.2 INTUTIONS AND DISCUSSIONS ABOUT THE IMPLICIT GENERALIZED POLICY IMPROVEMENT

The intuition for the expectile distillation loss (Eq. 7) is that the scalar Q function $Q(\cdot, \cdot)$ is a *one-step* summary of the average reward at future states sampled from the flow occupancy models, while the expectile loss serves as a "softmax" operator over the entire latent intention space. Theoretically, this expectile loss is guaranteed to converge to the maximum over $p(z)$ when $\mu \to 1$ (See Sec. 4.4 in Kostrikov et al. (2022) for details). Therefore, given an infinite amount of samples ($N \to \infty$) and an expectile $\mu \to 1$, the $Q$ converges to the greedy value functions:

$$Q^\star(s, a) = \max_{z \sim p(z)} \frac{1}{(1 - \gamma)} \mathbb{E}_{q_d(s_f \mid s, a, z)}[r(s_f)].$$

If we further assume that the flow occupancy models are optimal, i.e., $q_d^\star(s_f \mid s, a, z) = p^\beta(s_f \mid s, a, z)$, then the optimal $Q$ corresponds to a greedy value function under the behavioral policy $\beta$:

$$Q^\star(s, a) = \max_{z \sim p(z)} Q^\beta(s, a, z).$$

Unlike Q-learning, which converges to the optimal Q-function sequentially (Watkins & Dayan, 1992; Sutton et al., 1998), the implicit GPI proposes a new policy that is strictly no worse than the set of policies that correspond to each $Q_z$ in parallel (See Sec. 4.1 in Barreto et al. (2017) for further discussions). Unlike one-step policy improvement (Wang et al., 2018; Brandfonbrener et al., 2021; Peters & Schaal, 2007; Peters et al., 2010), implicit GPI can converge to the optimal policy for a downstream task, assuming that the task-specific intention has been captured during pre-training.

### C.3 ONE-STEP POLICY IMPROVEMENT WITH FLOW OCCUPANCY MODELS

The FOM + one-step PI variant performs one-step policy improvement using a flow occupancy model $q_d(s_f \mid s, a)$ that is *not* conditioned on latent intentions. This flow occupancy model captures the discounted state occupancy measure of the (average) behavioral policy. After training the flow occupancy model, FOM + one-step PI fits a Q function and extracts a behavioral-regularized policy:

$$Q \leftarrow \arg\min_Q \frac{1}{1 - \gamma} \mathbb{E}_{(s,a) \sim p^{\tilde\beta}(s,a), s_f \sim q_d(s_f \mid s, a)}[(Q(s, a) - r(s_f))^2],$$

$$\pi \leftarrow \arg\max_\pi \mathbb{E}_{(s,a) \sim p^{\tilde\beta}(s,a), a^\pi \sim \pi(a^\pi \mid s)} \left[ Q(s, a^\pi) + \alpha \log \pi(a \mid s) \right].$$

Intuitively, the first objective fits the behavioral Q function based on the dual definition (Eq. 2), and the second objective trains a policy to maximize this behavioral Q function, invoking one-step policy improvement. While this simple objective sometimes achieves strong performance on some benchmark tasks (Brandfonbrener et al., 2021; Eysenbach et al., 2022), it does not guarantee convergence to the optimal policy due to the use of a behavioral value function.

## D EXPERIMENTAL DETAILS

### D.1 TASKS AND DATASETS

Our experiments use a suite of 36 state-based and 4 image-based control tasks taken from ExORL benchmarks Yarats et al. (2022) and OGBench task suite (Park et al., 2025a) (Fig. 2).

**ExORL.** We use 16 state-based tasks from the ExORL (Yarats et al., 2022) benchmarks based on the DeepMind Control Suite (Tassa et al., 2018). These tasks involve controlling four robots (`cheetah`, `walker`, `quadruped`, and `jaco`) to achieve different locomotion behaviors. For each domain, the specific tasks are: `cheetah {run, run backward, walk, walk backward}`, `walker`

`{walk, run, stand, flip}`, quadruped `{run, jump, stand, walk}`, jaco `{reach top left,reach top right,reach bottom left,reach bottom right}`. For all tasks in `cheetah`, `walker`, and `quadruped`, both the episode length and the maximum return are 1000. For all tasks in `jaco`, both the episode length and the maximum return are 250. Following prior work (Park et al., 2024b), we multiply the return of `jaco` tasks by 4 to match other ExORL tasks during aggregation.

Following the prior work (Touati et al., 2023; Park et al., 2024b; Kim et al., 2024), we will use 5M unlabeled transitions collected by some exploration methods (e.g., RND (Burda et al., 2019)) for pre-training, and another 500K reward-labeled transitions collected by the same exploratory policy for fine-tuning. The fine-tuning datasets are labeled with task-specific dense rewards (Yarats et al., 2022), except in the `jaco` domains, where the reward signals are sparse.

**OGBench.** We use 20 state-based manipulation tasks from four domains (`cube single`, `cube double`, `scene`, and `puzzle 4x4`) in the OGBench task suite Park et al. (2025a), where the goal is to control a simulated robot arm to rearrange various objects. For each domain, the specific tasks are: `cube single` {task 1 (pick and place cube to left), `task 2` (pick and place cube to front), `task 3` (pick and place cube to back), `task 4` (pick and place cube diagonally), `task 5` (pick and place cube off-diagonally)}, `cube double` {`task 1` (pick and place one cube), `task 2` (pick and place two cubes to right), `task 3` (pick and place two cubes off-diagonally), `task 4` (swap cubes), `task 5` (stack cubes)}, `scene` {`task 1` (open drawer and window), `task 2` (close and lock drawer and window), `task 3` (open drawer, close window, and pick and place cube to right), `task 4` (put cube in drawer), `task 5` (fetch cube from drawer and close window)}, `puzzle 4x4` {`task 1` (all red to all blue), `task 2` (all blue to central red), `task 3` (two blue to mix), `task 4` (central red to all red), `task 5` (mix to all red)}. Note that some of these tasks, e.g., `cube double task 5` (stack cubes) and `scene task 4` (put cube in drawer), involve interacting with the environment in a specific order and thus require long-horizon temporal reasoning. For all tasks in `cube single`, `cube double`, and `scene`, the maximum episode length is 400. For all tasks in `puzzle 4x4`, the maximum episode length is 800. We also use 4 image-based tasks in the OGBench task suite. Specifically, we consider `visual cube single task 1`, `visual cube double task 1`, `visual scene task 1`, and `visual puzzle 4x4 task 1` from each domain, respectively. The observations are $64 \times 64 \times 3$ RGB images. These tasks are challenging because the agent needs to reason from pixels directly. All the manipulation tasks from OGBench are originally designed for evaluating goal-conditioned RL algorithms (Park et al., 2025a).

For both state-based and image-based tasks from OGBench, we will use 1M unlabeled transitions collected by a non-Markovian expert policy with temporally correlated noise (the `play` datasets) for pre-training, and another 500K reward-labeled transitions collected by the same noisy expert policy for fine-tuning. Unlike the ExORL benchmarks, the fine-tuning datasets for OGBench tasks are relabeled with *semi-sparse* rewards (Park et al., 2025b), providing less supervision for the algorithm.

## D.2 BASELINES

We compare InFOM with eight baselines across five categories of prior methods, focusing on different strategies for pre-training and fine-tuning in RL. First, implicit Q-Learning (IQL) (Kostrikov et al., 2022) and revisited behavior-regularized actor-critic (ReBRAC) (Tarasov et al., 2023a) are state-of-the-art offline RL algorithms based on the standard actor-critic framework (Appendix A.1). Second, we compare to a variant of ReBRAC learning on top of representations pre-trained on the unlabeled datasets. We chose an off-the-shelf self-supervised learning objective in vision tasks called self-distillation with no labels (DINO) (Caron et al., 2021) as our representation learning loss and name the resulting baseline DINO + ReBRAC. Third, our next baseline, model-based policy optimization (MBPO) (Janner et al., 2019), pre-trains a one-step model to predict transitions in the environment, similar to the next token prediction in language models (Radford et al., 2018). The one-step model is then used to augment the datasets for downstream policy optimization. We will again use ReBRAC to extract the policy (MBPO + ReBRAC). Fourth, we also include comparisons against the InfoNCE variant of contrastive RL (Eysenbach et al., 2019) and temporal difference InfoNCE (Zheng et al., 2024b), which pre-train the discounted state occupancy measure using Monte Carlo or temporal difference contrastive losses. While our method fits generative occupancy models, These two approaches predict the ratio of occupancy measures over some marginal densities serving

Table 1: Common hyperparameters for our method and the baselines.

| Hyperparameter | Value |
|---|---|
| learning rate | $3 \times 10^{-4}$ |
| optimizer | Adam (Kingma, 2014) |
| pre-training gradient steps | $1 \times 10^6$ for state-based tasks, $2.5 \times 10^5$ for image-based tasks |
| fine-tuning gradient steps | $5 \times 10^5$ for state-based tasks, $1 \times 10^5$ for image-based tasks |
| batch size | 256 |
| MLP hidden layer sizes | $(512, 512, 512, 512)$ |
| MLP activation function | GELU (Hendrycks & Gimpel, 2016) |
| discount factor $\gamma$ | 0.99 |
| target network update coefficient | $5 \times 10^{-3}$ |
| double Q aggregation | min |
| policy update frequency in fine-tuning | 1/4 |
| image encoder | small IMPALA encoder (Espeholt et al., 2018; Park et al., 2025b) |
| image augmentation method | random cropping |
| image augmentation probability | 1.0 for DINO + ReBRAC, 0.5 for all other methods |
| image frame stack | 3 |

as the discriminative counterparts. After pre-training the ratio predictors, importance sampling is required to recover the Q function (CRL + IS & TD InfoNCE + IS) (Mazoure et al., 2023; Zheng et al., 2024b), enabling policy maximization. Fifth, our final set of baselines are prior unsupervised RL methods that pre-train a set of latent intentions and intention-conditioned policies using forward-backward representations (Touati & Ollivier, 2021) or a Hilbert space (Park et al., 2024b). Given a downstream task, these methods first infer the corresponding intention in a zero-shot manner and then fine-tune the policy using offline RL (Kim et al., 2024), differing from the implicit GPI as in our method. We will use IQL as the fine-tuning algorithm and call the resulting methods FB + IQL and HILP + IQL. For image-based tasks, we selectively compare to four baselines: ReBRAC, CRL + IS, DINO + ReBRAC, and FB + IQL.

## D.3 EVALUATION PROTOCOLS

We compare the performance of InFOM against the eight baselines (Sec. 5.1) after first pre-training each method for 1M gradient steps (250K gradient steps for image-based tasks) and then fine-tuning for 500K gradient steps (100K gradient steps for image-based tasks). We measure the episode return for tasks from ExORL benchmarks and the success rate for tasks from the OGBench task suite. For OGBench tasks, the algorithms still use the semi-sparse reward instead of the success rate for training. Following prior practice (Park et al., 2025b; Tarasov et al., 2023b), we do *not* report the best performance during fine-tuning and report the evaluation results averaged over 400K, 450K, and 500K gradient steps instead. For image-based tasks, we report the evaluation results averaged over 50K, 75K, and 100K gradient steps during fine-tuning. For evaluating the performance of different methods throughout the entire fine-tuning process, we defer the details to specific figures (e.g., Fig. 10 & 9).

## D.4 IMPLEMENTATIONS AND HYPERPARAMETERS

In this section, we discuss the implementation details and hyperparameters for InFOM and the eight baselines. Whenever possible, we use the same set of hyperparameters for all methods (Table 1) across all tasks, including learning rate, network architecture, batch size, image encoder, etc. Of particular note is that we use asynchronous policy training (Zhou et al., 2025), where we update the policy 4 times less frequently than other models during fine-tuning. For specific hyperparameters of each method, we tune them on the following tasks from each domain and use one set of hyperparameters for every task in that domain. For image-based tasks, we tune hyperparameters for each task individually.

- `cheetah`: cheetah run
- `walker`: walker walk
- `quadruped`: quadruped jump

Table 2: **Hyperparameters for InFOM.** See Appendix D.4 for descriptions of each hyperparameter.

| Hyperparameter | Value |
|---|---|
| latent intention dimension $d$ | See Table 3 |
| number of steps for the Euler method $T$ | 10 |
| number of future states $N$ | 16 |
| normalize the Q loss term in $\mathcal{L}(\pi)$ (Eq. 8) | No |
| expectile $\mu$ | See Table 3 |
| KL divergence regularization coefficient $\lambda$ | See Table 3 |
| behavioral cloning regurlaization coefficient $\alpha$ | See Table 3 |

Table 3: **Domain-specific hyperparameters for our method and the baselines.** We individually tune these hyperparameters for each domain and use the same set of hyperparameters for tasks in the same domain. See Appendix D.4 for tasks used to tune these hyperparameters and descriptions of each hyperparameter. "-" indicates that the hyperparameter does not exist.

| Domain or Task | InFOM (Ours) | | | | IQL | ReBRAC | | DINO + ReBRAC | MBPO + ReBRAC | | CRL + IS | TD InfoNCE + IS | FB + IQL | | HILP + IQL |
|---|---|---|---|---|---|---|---|---|---|---|---|---|---|---|---|
| | $d$ | $\mu$ | $\lambda$ | $\alpha$ | $\alpha$ | $\alpha_{\text{actor}}$ | $\alpha_{\text{critic}}$ | $\kappa_{\text{student}}$ | $N_{\text{imaginary}}$ | $H_{\text{imaginary}}$ | $\alpha$ | $\alpha$ | $\alpha_{\text{repr}}$ | $\alpha_{\text{AWR}}$ | $\alpha$ |
| cheetah | 128 | 0.9 | 0.05 | 0.3 | 1 | 0.1 | 0.1 | 0.1 | 128 | 1 | 0.03 | 0.003 | 1 | 1 | 1 |
| walker | 512 | 0.9 | 0.1 | 0.3 | 1 | 10 | 0.1 | 0.1 | 128 | 1 | 0.03 | 0.03 | 1 | 10 | 10 |
| quadruped | 512 | 0.9 | 0.005 | 0.3 | 10 | 1 | 1 | 0.1 | 128 | 1 | 0.03 | 0.03 | 10 | 1 | 10 |
| jaco | 512 | 0.9 | 0.2 | 0.1 | 0.1 | 0.1 | 0.1 | 0.1 | 128 | 1 | 0.003 | 0.03 | 1 | 1 | 1 |
| cube single | 512 | 0.95 | 0.05 | 30 | 1 | 1 | 1 | 0.04 | 256 | 2 | 30 | 30 | 10 | 1 | 1 |
| cube double | 128 | 0.9 | 0.025 | 30 | 1 | 1 | 1 | 0.04 | 256 | 2 | 30 | 30 | 1 | 10 | 1 |
| scene | 128 | 0.99 | 0.2 | 300 | 1 | 1 | 1 | 0.1 | 256 | 2 | 3 | 3 | 10 | 10 | 1 |
| puzzle 4x4 | 128 | 0.95 | 0.1 | 300 | 10 | 0.1 | 0.1 | 0.1 | 256 | 2 | 3 | 3 | 10 | 10 | 1 |
| visual cube single task 1 | 512 | 0.95 | 0.025 | 30 | - | 1 | 0 | 0.1 | - | - | 30 | - | 10 | 1 | - |
| visual cube double task 1 | 128 | 0.9 | 0.01 | 30 | - | 0.1 | 0 | 0.1 | - | - | 30 | - | 10 | 1 | - |
| visual scene task 1 | 128 | 0.99 | 0.1 | 300 | - | 0.1 | 0.01 | 0.1 | - | - | 3 | - | 10 | 10 | - |
| visual puzzle 4x4 task 1 | 128 | 0.95 | 0.1 | 300 | - | 0.1 | 0.01 | 0.1 | - | - | 3 | - | 10 | 10 | - |

- jaco: jaco reach top left
- cube single: cube single task 2
- cube double: cube double task 2
- scene: scene task 2
- puzzle 4x4: puzzle 4x4 task 4

**InFOM.** InFOM consists of two main components for pre-training: the intention encoder and the flow occupancy models. First, we use a Gaussian distribution conditioned on the next state-action pair as the intention encoding distribution. Following prior work (Kingma & Welling, 2013; Alemi et al., 2017), we model the intention encoder as a multilayer perceptron (MLP) that takes the next state-action pair $(s', a')$ as input and outputs two heads representing the mean and the (log) standard deviation of the Gaussian. We apply layer normalization to the intention encoder to stabilize optimization. We use the reparameterization trick (Kingma & Welling, 2013) to backpropagate the gradients from the flow-matching loss and the KL divergence regularization (Eq. 4) into the intention encoder. Our initial experiments suggest that the dimension of the latent intention space $d$ is an important hyperparameter, and we sweep over $\{64, 128, 256, 512\}$ and find that $d = 512$ is sufficient for most ExORL tasks and $d = 128$ is generally good enough for all OGBench tasks. For the coefficient of the KL divergence regularization $\lambda$, we sweep over $\{2.0, 1.0, 0.2, 0.1, 0.05, 0.025, 0.01, 0.005\}$ to find the best $\lambda$ for each domain. Second, we use flow-matching vector fields to model the flow occupancy models. The vector field is an MLP that takes in a noisy future state $s_f^t$, a state-action pair $(s, a)$, and a latent intention $z$, and outputs the vector field with the same dimension as the state. We apply layer normalization to the vector field to stabilize optimization. As mentioned in Sec. 3, we use flow-matching objectives based on optimal transport (linear path) and sample the time step $t$ from the uniform distribution over the unit interval. Following prior work (Park et al., 2025b), we use a fixed $T = 10$ steps (step size = 0.1) for the Euler method and do not apply a sinusoidal embedding for the time. To make a fair comparison with other baselines, we also pre-train a behavioral cloning policy that serves as initialization for fine-tuning.

For fine-tuning, InFOM learns three components: the reward predictor, the critic, and the policy, while fine-tuning the intention encoder and the flow occupancy models. The reward predictor is an

MLP that predicts the scalar reward of a state trained using mean squared error. We apply layer normalization to the reward predictor to stabilize learning. The critic is an MLP that predicts double Q values (Van Hasselt et al., 2016; Fujimoto et al., 2018) of a state-action pair, without conditioning on the latent intention. We apply layer normalization to the critic to stabilize learning. We train the critic using the expectile distillation loss (Eq. 7) and sweep the expectile over $\{0.9, 0.95, 0.99\}$ to find the best $\mu$ for each domain. We use $N = 16$ future states sampled from the flow occupancy models to compute the average reward, which we find to be sufficient. We use the minimum of the double Q predictions to prevent overestimation. The policy is an MLP that outputs a Gaussian distribution with a unit standard deviation. In our initial experiments, we find that the behavioral cloning coefficient $\alpha$ in Eq. 8 is important, and we sweep over $\{300, 30, 3, 0.3\}$ to find the best $\alpha$ for each domain. Following prior practice (Park et al., 2025b), we do not normalize the Q loss term in the actor loss $\mathcal{L}(\pi)$ (Eq. 8) as in Fujimoto & Gu (2021). Other choices of the policy network include the diffusion model (Ren et al., 2025; Wang et al., 2023) and the flow-matching model (Park et al., 2025b), and we leave investigating these policy networks to future work.

For image-based tasks, following prior work (Park et al., 2025b), we use a smaller variant of the IMPALA encoder (Espeholt et al., 2018) and apply random cropping augmentation with a probability of 0.5. We also apply frame stacking with three images. Table 2 and Table 3 summarize the hyperparameters for InFOM.

**IQL and ReBRAC.** We reuse the IQL (Kostrikov et al., 2022) implementation and the Re-BRAC (Tarasov et al., 2023a) implementation from Park et al. (2025b). Since learning a critic requires reward-labeled datasets or relabeling rewards for unlabeled datasets (Yu et al., 2022), we simply pre-train a behavioral cloning policy. During the fine-tuning, we use the behavioral cloning policy as initialization and train a critic from scratch using the TD error (Kostrikov et al., 2022; Fujimoto & Gu, 2021; Tarasov et al., 2023a). Following prior work (Park et al., 2025b), we use the same expectile value 0.9 for IQL on all tasks, and sweep over $\{100, 10, 1, 0.1, 0.01\}$ to find the best AWR inverse temperature $\alpha$ for each domain. For ReBRAC, we tune the behavioral cloning (BC) regularization coefficients for the actor and the critic separately. We use the range $\{100, 10, 1, 0.1\}$ to search for the best actor BC coefficient $\alpha_{\text{actor}}$ and use the range $\{100, 10, 1, 0.1, 0\}$ to search for the best critic BC coefficient $\alpha_{\text{critic}}$. We use the default values for other hyperparameters following the implementation from Park et al. (2025b). See Table 3 for domain-specific hyperparameters.

**DINO + ReBRAC.** We implement DINO on top of ReBRAC. DINO (Caron et al., 2021) learns a state encoder using two augmentations of the same state. For state-based tasks, the state encoder is an MLP that outputs representations. We apply two clipped Gaussian noises centered at zero to the same state to obtain those augmentations. The standard deviation of the Gaussian noise is set to 0.2, and we clip the noise into $[-0.2, 0.2]$ on all domains. For image-based tasks, the state encoder is the small IMPALA encoder that also outputs representations. We apply two different random cropings to the same image observation to obtain those augmentations. We sweep over $\{0.01, 0.04, 0.1, 0.4\}$ for the temperature for student representations $\kappa_{\text{student}}$ and use a fixed temperature 0.04 for teacher representations on all domains. We use a representation space with 512 dimensions. We update the target representation centroid with a fixed ratio of 0.1. During pre-training, we learns the DINO representations along with a behavioral cloning policy. During fine-tuning, we learn the actor and the critic using ReBRAC on top of DINO representations, while continuing to fine-tune those DINO representations. We use the same BC coefficients $\alpha_{\text{actor}}$ and $\alpha_{\text{critic}}$ as in ReBRAC. For image-based tasks, we apply random cropping to the same image twice with a probability of 1.0 and use those two augmentations to compute the teacher and the student representations. See Table 3 for domain-specific temperatures for student representations.

**MBPO + ReBRAC.** We implement MBPO (Janner et al., 2019) on top of ReBRAC and only consider this baseline for state-based tasks. MBPO learns a one-step transition MLP to predict the residual between the next state $s'$ and the current state $s$ conditioned on the current state-action pair $(s, a)$. We pre-train the one-step model with a behavioral cloning policy. During fine-tuning, we use the model with a learned reward predictor to collect imaginary rollouts. We *only* use these imaginary rollouts to learn the actor and the critic. We sweep over $\{64, 128, 256\}$ for the number of imaginary rollouts to collect for each gradient step $N_{\text{imaginary}}$ and sweep over $\{1, 2, 4\}$ for the number of steps in each rollout $H_{\text{imaginary}}$. We use the same BC coefficient as in ReBRAC. See Table 3 for the domain-specific number of imaginary rollouts and number of steps in each rollout.

**CRL + IS and TD InfoNCE + IS.** We mostly reuse the CRL (Eysenbach et al., 2022) implementation based on the InfoNCE loss from Park et al. (2025a) and adapt it to our setting by adding the important sampling component. We implement TD InfoNCE by adapting the official implementations (Zheng et al., 2024b). For both methods, we pre-train the classifiers that predict the ratio between the occupancy measures and the marginal densities over future states with a behavioral cloning policy. We use the SARSA variant of TD InfoNCE during pre-training. After pre-training the classifiers, we learn a reward predictor and apply importance sampling weights predicted by the classifiers to a set of future states sampled from the fine-tuning datasets to estimate $Q$. This $Q$ estimation then drives policy optimization. We use a single future state from the fine-tuning dataset to construct the importance sampling estimation, which is sufficient. We use 512-dimensional contrastive representations. We sweep over $\{300, 30, 3, 0.3, 0.03\}$ for the BC coefficient $\alpha$ (Table 3).

**FB + IQL and HILP + IQL.** We implement FB (Touati & Ollivier, 2021) and HILP (Park et al., 2024b) by adapting the FB implementation from Jeen et al. (2024) and the HILP implementation from Kim et al. (2024). During pre-training, for FB, we pre-train the forward-backward representations and the intention-conditioned policies in an actor-critic manner. We use a coefficient 1 for the orthonormality regularization of the backward representations. We use 512-dimensional forward-backward representations. We sample the latent intentions for pre-training from either a standard Gaussian distribution (with probability 0.5) or the backward representations for a batch of states (with probability 0.5), normalizing those latent intentions to length $\sqrt{512}$. We sweep over $\{100, 10, 1, 0.1\}$ for the BC coefficient $\alpha_{\text{repr}}$. For HILP, we pre-train the Hilbert representations $\phi$ and Hilbert foundation policies using an actor-critic framework as well. We use implicit value learning to learn the Hilbert representations following implementations from Park et al. (2024a; 2025a). We set the expectile to 0.9 for all domains. We sweep over $\{100, 10, 1, 0.1\}$ to find the best AWR inverse temperature $\alpha$. We also use a 512-dimensional Hilbert representation space. To construct the intrinsic rewards, we first sample the latent intention $z$ from a standard Gaussian, normalizing them to length $\sqrt{512}$, and then use the representation of the next state $\phi(s')$ and the representation of the current state $\phi(s)$ to compute the intrinsic reward $(\phi(s') - \phi(s))^\top z$.

During fine-tuning, we first infer a task-specific backward representation or a Hilbert representation using a small amount of transitions (10K) from the fine-tuning datasets, and then invoke IQL to learn the critic and the actor using downstream rewards conditioned on the inferred representations. For FB, we sweep over $\{100, 10, 1, 0.1\}$ for the AWR inverse temperature $\alpha_{\text{AWR}}$ for IQL. For HILP, we reuse the same AWR inverse temperature in representation learning for IQL. See Table 3 for domain-specific BC coefficients and AWR inverse temperatures.

# E   ADDITIONAL VISUALIZATIONS OF LATENT INTENTIONS

We include additional visualization of latent intentions on `quadruped-jump` in Fig. 6.

# F   ADDITIONAL EXPERIMENTS

## F.1   EVALUATION ON ROBOTICS BENCHMARKS

To further study the pre-training and fine-tuning effects of our method on realistic datasets. Specifically, we choose the RT-1 dataset (Brohan et al., 2022), which contains 73499 episodes of transitions. This dataset was collected by commanding a Google robot to pick, place, and move 17 objects in the Google micro-kitchens, covering a diverse set of intentions. Since collecting distinct robotics datasets for pre-training and fine-tuning is difficult, we use the entire dataset as both the reward-free pre-training dataset and the reward-labeled fine-tuning dataset. For the evaluation task, we use `google robot pick coke can` from the SimplerEnv (Li et al., 2024), which contains a suite of simulation tasks that efficiently and informatively complement real-world evaluations of the Google robot.

We compare against two baselines from our experiments (ReBRAC and DINO + ReBRAC) due to computational constraints, and also include a behavioral cloning (BC) baseline for reference. Our initial experiments indicate that all the algorithms (except DINO + ReBRAC) perform poorly when trained end-to-end from pixels directly. Following prior practice in latent flow matching (Rombach

Table 4: **Evaluation on ExORL and OGBench benchmarks.** Following OGBench (Park et al., 2025a), we bold values at and above 95% of the best performance for each task.

| Task | InFOM (Ours) | IQL | ReBRAC | DINO + ReBRAC | MBPO + ReBRAC | CRL + IS | TD InfoNCE + IS | FB + IQL | HILP + IQL |
|---|---|---|---|---|---|---|---|---|---|
| cheetah run | 97.6 ± 7.8 | 80.0 ± 8.4 | 97.2 ± 12.9 | 87.2 ± 8.6 | **104.7 ± 2.4** | 73.3 ± 6.7 | 68.2 ± 8.9 | 83.3 ± 10.9 | 90.3 ± 1.9 |
| cheetah run backward | **104.7 ± 7.3** | 77.0 ± 12.6 | 84.9 ± 3.7 | 67.1 ± 6.4 | 87.0 ± 4.8 | 74.7 ± 8.1 | 74.3 ± 17.1 | 67.3 ± 7.0 | 64.4 ± 6.4 |
| cheetah walk | 254.8 ± 158.6 | 357.9 ± 16.4 | **443.4 ± 15.3** | 383.5 ± 10.3 | **447.4 ± 12.7** | 327.4 ± 38.7 | 336.7 ± 22.1 | 346.5 ± 24.3 | 366.8 ± 6.9 |
| cheetah walk backward | 251.8 ± 116.9 | 303.7 ± 12.6 | **403.0 ± 16.1** | 318.4 ± 23.0 | **398.6 ± 16.0** | 330.2 ± 8.5 | 326.3 ± 45.1 | 298.0 ± 22.8 | 318.1 ± 11.4 |
| walker walk | **467.3 ± 82.1** | 208.6 ± 3.7 | 208.1 ± 5.8 | 228.0 ± 3.7 | 327.6 ± 4.5 | 213.3 ± 7.8 | 212.2 ± 13.2 | 225.3 ± 6.7 | 225.4 ± 3.7 |
| walker run | **116.3 ± 15.3** | 92.4 ± 0.6 | 97.8 ± 1.2 | 98.5 ± 1.0 | 107.6 ± 1.2 | 91.5 ± 3.2 | 91.0 ± 3.7 | 97.4 ± 1.2 | 97.4 ± 2.2 |
| walker stand | **581.2 ± 72.1** | 409.1 ± 2.3 | 460.6 ± 1.1 | 453.0 ± 3.1 | 458.1 ± 2.5 | 409.0 ± 7.5 | 397.2 ± 6.0 | 446.8 ± 7.1 | 443.3 ± 3.8 |
| walker flip | **358.8 ± 10.3** | 260.3 ± 2.8 | **344.6 ± 2.7** | 320.3 ± 4.3 | **341.8 ± 3.7** | 255.0 ± 8.0 | 231.6 ± 6.9 | 287.0 ± 3.1 | 280.7 ± 5.4 |
| quadruped run | 341.8 ± 41.2 | 358.0 ± 6.2 | 343.0 ± 2.6 | 344.7 ± 2.9 | **395.1 ± 2.6** | 323.4 ± 2.9 | 222.1 ± 39.7 | 367.0 ± 3.8 | 371.1 ± 11.5 |
| quadruped jump | 626.0 ± 6.8 | 628.5 ± 7.8 | 605.2 ± 7.8 | 573.0 ± 9.6 | **666.9 ± 3.4** | 576.7 ± 13.7 | 421.4 ± 93.4 | **639.4 ± 8.9** | 626.5 ± 14.5 |
| quadruped stand | **718.3 ± 18.7** | **714.2 ± 9.8** | 688.6 ± 5.0 | 663.2 ± 8.3 | 703.7 ± 3.6 | 653.1 ± 8.4 | 457.1 ± 47.7 | **728.9 ± 11.5** | **715.6 ± 13.9** |
| quadruped walk | 360.7 ± 7.9 | **375.1 ± 3.7** | 343.5 ± 7.1 | **391.4 ± 7.2** | **390.0 ± 5.7** | 309.6 ± 9.6 | 243.1 ± 29.2 | **388.9 ± 7.0** | **393.4 ± 3.4** |
| jaco reach top left | **742.5 ± 43.7** | 74.7 ± 19.6 | 59.0 ± 4.9 | 17.5 ± 3.8 | 60.1 ± 6.2 | 29.1 ± 4.7 | 31.5 ± 3.0 | 25.0 ± 11.4 | 40.4 ± 11.5 |
| jaco reach top right | **687.5 ± 46.7** | 40.6 ± 14.0 | 38.0 ± 13.1 | 11.0 ± 4.1 | 52.5 ± 10.8 | 21.4 ± 6.5 | 25.5 ± 10.3 | 16.2 ± 3.2 | 25.1 ± 9.6 |
| jaco reach bottom left | **746.7 ± 12.6** | 77.1 ± 12.5 | 44.5 ± 4.0 | 13.7 ± 2.8 | 43.4 ± 4.6 | 19.8 ± 8.8 | 26.6 ± 5.9 | 19.8 ± 4.0 | 27.8 ± 4.6 |
| jaco reach bottom right | **733.0 ± 19.6** | 78.7 ± 19.1 | 41.4 ± 5.0 | 8.3 ± 2.8 | 34.0 ± 6.0 | 19.6 ± 2.0 | 25.4 ± 5.7 | 12.4 ± 2.7 | 24.7 ± 3.9 |
| cube single task 1 | **92.5 ± 4.0** | 53.0 ± 8.7 | 67.3 ± 14.2 | 1.8 ± 1.0 | 77.8 ± 11.7 | 10.1 ± 2.7 | 13.8 ± 3.8 | 17.7 ± 8.8 | 32.9 ± 9.2 |
| cube single task 2 | 78.4 ± 12.3 | 51.7 ± 15.1 | **93.7 ± 3.5** | 1.2 ± 0.6 | **94.2 ± 2.0** | 3.7 ± 2.8 | 8.5 ± 5.6 | 16.7 ± 8.6 | 26.5 ± 15.4 |
| cube single task 3 | 56.4 ± 36.9 | 41.5 ± 5.3 | **94.8 ± 0.8** | 1.5 ± 1.4 | **93.1 ± 4.7** | 12.5 ± 3.2 | 11.7 ± 7.4 | 16.0 ± 12.2 | 35.5 ± 14.7 |
| cube single task 4 | **91.5 ± 14.2** | 42.2 ± 8.3 | **89.5 ± 3.6** | 0.5 ± 1.0 | **88.7 ± 4.7** | 1.7 ± 1.7 | 3.3 ± 3.0 | 18.7 ± 9.9 | 36.4 ± 14.9 |
| cube single task 5 | 70.0 ± 39.1 | 33.7 ± 12.9 | 83.3 ± 6.8 | 0.5 ± 0.6 | **87.8 ± 2.7** | 4.3 ± 2.2 | 4.0 ± 3.2 | 14.2 ± 12.0 | 18.5 ± 5.6 |
| cube double task 1 | **29.3 ± 10.5** | 17.8 ± 9.6 | 2.2 ± 1.7 | 0.0 ± 0.0 | 2.7 ± 1.1 | 4.1 ± 1.9 | 6.7 ± 2.7 | 0.2 ± 0.3 | 0.7 ± 1.1 |
| cube double task 2 | **12.5 ± 10.7** | 1.3 ± 1.2 | 0.0 ± 0.0 | 0.0 ± 0.0 | 0.0 ± 0.0 | 0.0 ± 0.0 | 0.0 ± 0.0 | 0.0 ± 0.0 | 0.0 ± 0.0 |
| cube double task 3 | **11.6 ± 8.3** | 0.3 ± 0.4 | 0.0 ± 0.0 | 0.0 ± 0.0 | 0.0 ± 0.0 | 0.0 ± 0.0 | 0.0 ± 0.0 | 0.0 ± 0.0 | 0.0 ± 0.0 |
| cube double task 4 | **0.3 ± 0.4** | 0.0 ± 0.0 | 0.0 ± 0.0 | 0.0 ± 0.0 | 0.0 ± 0.0 | 0.0 ± 0.0 | 0.0 ± 0.0 | 0.0 ± 0.0 | 0.0 ± 0.0 |
| cube double task 5 | **2.8 ± 4.6** | 1.5 ± 1.0 | 0.0 ± 0.0 | 0.0 ± 0.0 | 0.0 ± 0.0 | 0.0 ± 0.0 | 0.3 ± 0.7 | 0.0 ± 0.0 | 0.0 ± 0.0 |
| scene task 1 | **97.8 ± 1.0** | 66.5 ± 13.1 | 47.7 ± 7.2 | 26.7 ± 4.3 | 35.3 ± 7.7 | 17.5 ± 5.1 | 21.0 ± 4.3 | 12.3 ± 11.3 | 8.8 ± 3.0 |
| scene task 2 | **15.6 ± 3.4** | 2.5 ± 1.5 | 7.8 ± 4.9 | 1.3 ± 0.0 | 5.6 ± 5.6 | 2.3 ± 0.7 | 1.7 ± 1.3 | 1.5 ± 1.8 | 1.2 ± 1.7 |
| scene task 3 | **43.5 ± 2.8** | 0.7 ± 0.5 | 1.7 ± 1.1 | 0.2 ± 0.3 | 2.4 ± 0.8 | 0.8 ± 0.3 | 0.5 ± 1.0 | 0.0 ± 0.0 | 0.0 ± 0.0 |
| scene task 4 | 1.0 ± 0.7 | 0.2 ± 0.3 | **2.8 ± 0.8** | 0.2 ± 0.3 | 2.0 ± 1.3 | 1.2 ± 1.4 | 0.7 ± 1.3 | 0.2 ± 0.3 | 0.0 ± 0.0 |
| scene task 5 | **0.3 ± 0.4** | 0.0 ± 0.0 | 0.0 ± 0.0 | 0.0 ± 0.0 | 0.0 ± 0.0 | **0.3 ± 0.1** | 0.0 ± 0.0 | 0.0 ± 0.0 | 0.0 ± 0.0 |
| puzzle 4x4 task 1 | **24.2 ± 14.4** | 2.3 ± 2.3 | 12.8 ± 3.1 | 0.3 ± 0.7 | 16.9 ± 1.4 | 0.0 ± 0.0 | 0.0 ± 0.0 | 0.2 ± 0.3 | 0.3 ± 0.6 |
| puzzle 4x4 task 2 | **14.5 ± 9.4** | 0.5 ± 0.6 | 0.5 ± 0.6 | 0.0 ± 0.0 | 0.2 ± 0.4 | 0.3 ± 0.4 | 0.0 ± 0.0 | 0.2 ± 0.3 | 0.4 ± 0.6 |
| puzzle 4x4 task 3 | **26.3 ± 13.4** | 1.0 ± 0.9 | 5.0 ± 2.7 | 0.0 ± 0.0 | 5.1 ± 2.8 | 0.3 ± 0.4 | 0.0 ± 0.0 | 0.2 ± 0.3 | 0.1 ± 0.3 |
| puzzle 4x4 task 4 | **12.0 ± 7.1** | 0.3 ± 0.7 | 0.8 ± 0.8 | 0.0 ± 0.0 | 0.4 ± 0.4 | 0.0 ± 0.0 | 0.0 ± 0.0 | 0.2 ± 0.3 | 0.1 ± 0.3 |
| puzzle 4x4 task 5 | **12.3 ± 6.2** | 0.7 ± 0.8 | 0.0 ± 0.0 | 0.0 ± 0.0 | 0.0 ± 0.0 | 0.5 ± 0.6 | 0.0 ± 0.0 | 0.0 ± 0.0 | 0.1 ± 0.3 |
| visual cube single task 1 | **52.1 ± 20.8** | - | 10.6 ± 7.2 | 15.3 ± 14.6 | - | 12.0 ± 5.6 | - | 31.0 ± 15.0 | - |
| visual cube double task 1 | **11.2 ± 9.2** | - | 0.0 ± 0.0 | 5.0 ± 2.0 | - | 5.0 ± 3.6 | - | 1.3 ± 1.5 | - |
| visual scene task 1 | 72.4 ± 17.7 | - | 32.0 ± 13.0 | 26.0 ± 17.2 | - | 9.0 ± 6.6 | - | **74.7 ± 22.2** | - |
| visual puzzle 4x4 task 1 | **6.0 ± 3.2** | - | 0.0 ± 0.0 | 0.0 ± 0.0 | - | 0.0 ± 0.0 | - | 0.0 ± 0.0 | - |

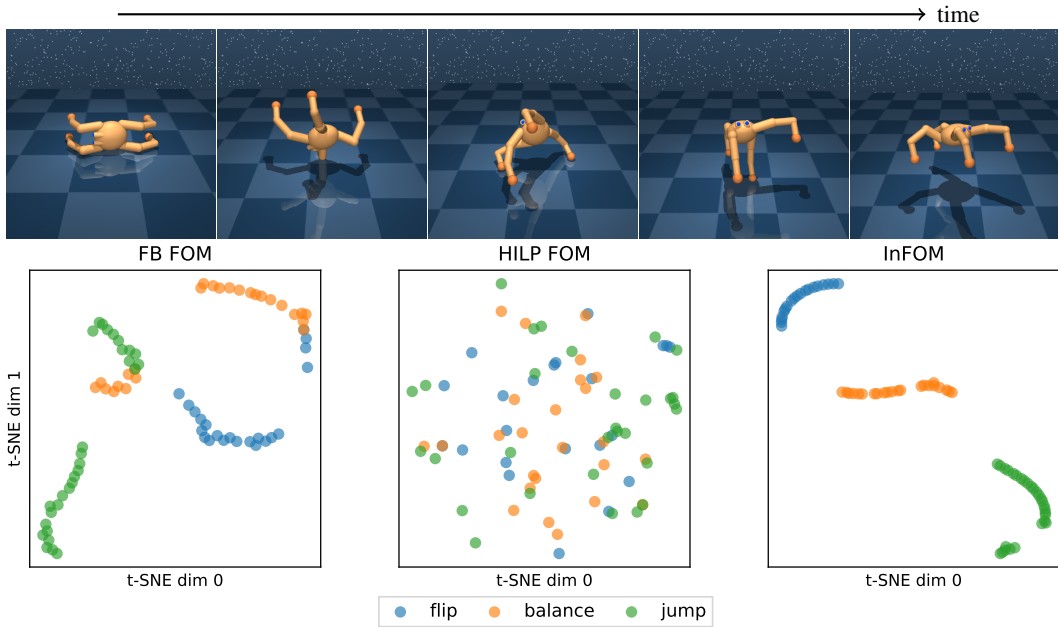

Figure 6: Visualization of latent intentions on `quadruped-jump`.

et al., 2022; Dao et al., 2023), we therefore pre-train a $\beta$-VAE (Higgins et al., 2017) to encode images into a latent embedding space and then learn algorithms on top of those embeddings. For DINO + ReBRAC, we directly use the image representations learned by DINO to train the actor and the critic. We report means and standard deviations of success rates over 4 random seeds.

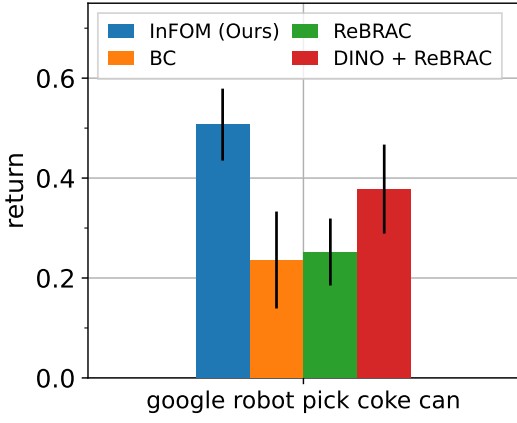 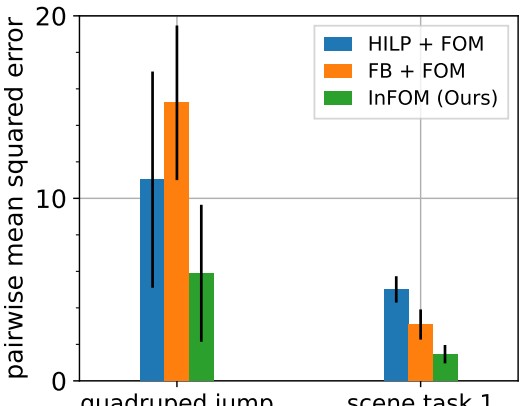

Figure 7: **Evaluation on robotics datasets.** InFOM outperforms the best baseline by 34% when trained on top of embeddings from a fixed image encoder. See Appendix F.1 for details.

Figure 8: **Comparison to prior intention encoding mechanisms after pre-training.** We compare InFOM to prior intention encoding mechanisms based on unsupervised skill discovery (HILP (Park et al., 2024b)) or successor feature learning (FB (Touati & Ollivier, 2021)) after pre-training. FB + FOM is equivalent to TD flows with GPI in Farebrother et al. (2025). InFOM achieves lower prediction errors on both tasks.

Results in Fig. 7 suggest that InFOM outperforms the best baseline by 34% when trained on top of embeddings from a fixed image encoder, indicating that our method can effectively fine-tune on challenging, realistic datasets with overlapping intentions.

## F.2 VARIATIONAL INTENTION INFERENCE IS SIMPLE AND PERFORMANT

We now conduct experiments ablating a key component in our method: the variational intention encoder. To investigate whether this framework induces a simple and performant way to infer diverse user intentions from an unlabeled dataset, we compare it to various intention encoding mechanisms proposed by prior methods. Specifically, we consider replacing the variational intention encoder with either *(1)* a set of Hilbert representations and Hilbert foundation policies (Park et al., 2024b) (HILP + FOM) or *(2)* a set of forward-backward representations and representation-conditioned policies (Touati & Ollivier, 2021) (FB + FOM), and then pre-training the flow occupancy models conditioned on these two sets of representations. Note that FB + FOM is equivalent to TD flows with GPI in Farebrother et al. (2025).

We first compare the future state predictions from InFOM against HILP + FOM and FB + FOM on two ExORL tasks (`quadruped jump` and `scene task 1`) after pre-training. Specifically, we compute the pairwise mean squared error (MSE) between predicted future states and ground-truth future states along a trajectory. We first sample 100 trajectories from the pre-training datasets, and then, for each trajectory, we sample 400 future states from InFOM and the two baselines starting from the same initial $(s, a)$ pair. We compute the pairwise MSE between each sampled future state and the corresponding sequence of ground-truth future states within the same trajectory. The prediction error is reported as the pairwise MSE averaged over all transitions in the 100 trajectories and the 400 sampled future states. Results in Fig. 8 show that InFOM achieves lower prediction errors than two FOM baselines.

We then compare the performance of InFOM against HILP + FOM and FB + FOM after fine-tuning. We choose two tasks in the ExORL benchmarks (`walker flip` and `quadruped jump`) and another two tasks taken from the OGBench benchmarks (`cube double task 1` and `scene task 1`), following the same evaluation protocols as in Appendix D.3. Results in Fig. 9 indicate that InFOM can outperform prior intention encoding methods on 3 of 4 tasks, while being simpler. Both HILP and FB capture intentions with full unsupervised RL objectives based on an actor-critic backbone. In contrast, we capture intentions by simply training an intention encoder based on a latent

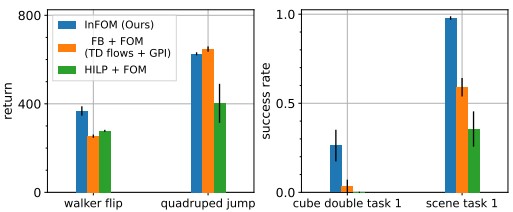 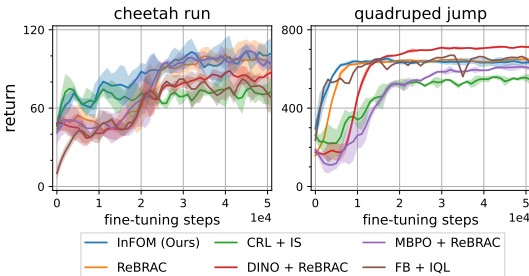

Figure 9: **Comparison to prior intention encoding mechanisms after fine-tuning.** We compare InFOM to prior intention encoding mechanisms based after fine-tuning. We observe that InFOM outperforms prior methods on 3 out of the 4 tasks.

Figure 10: **Convergence speed during fine-tuning.** On tasks where InFOM and baselines perform similarly, our flow occupancy models enable faster policy learning.

variable model over adjacent transitions, without relying on a potentially complicated offline RL procedure (Tarasov et al., 2023b; Park et al., 2024a).

### F.3 FLOW OCCUPANCY MODELS ENABLE FASTER POLICY LEARNING

We then investigate whether the proposed method leads to faster policy learning on downstream tasks. We answer this question by an ablation study with a high evaluation frequency, analyzing the performance of various methods throughout the entire fine-tuning phase every 2K gradient steps. We compare InFOM to prior methods on two ExORL tasks (`cheetah run` and `quadruped jump`), including ReBRAC, CRL + IS, DINO + ReBRAC, MBPO + ReBRAC, and FB + IQL (See Appendix D.2 for details of these baselines). We choose these baselines because they perform similarly to our method, helping to prevent counterfactual errors derived from the performance deviation when comparing convergence speed.

We compare different algorithms by plotting the returns at each evaluation step, with the shaded regions indicating one standard deviation. As shown in Fig. 10, InFOM converges faster than prior methods that only pre-train behavioral cloning policies (ReBRAC) or self-supervised state representations (DINO + ReBRAC), demonstrating the effectiveness of extracting temporal information. The observation that methods utilizing a one-step transition model (MBPO + ReBRAC) or a future state classifier (CRL + IS) learn more slowly than our method highlights the importance of predicting long-horizon future events using expressive generative models. Additionally, our flow occupancy models extract rich latent intentions from the unlabeled datasets, resulting in adaptation speed similar to the prior zero-short RL method (FB + IQL).

### F.4 LEARNING WITH DISCRETE INTENTIONS

The choice of the prior over latent variables $p(z)$ is still an open question in the literature. Prior work has used a standard Gaussian distribution (Frans et al., 2024), a uniform von Mises–Fisher distribution Park et al. (2024b); Touati & Ollivier (2021); Zheng et al. (2025), a continuous uniform distribution (Sharma et al., 2019), and a discrete uniform distribution (Eysenbach et al., 2019).

To further investigate the effect of using a discrete set of latent intentions for InFOM, we run additional ablation experiments. We selected a set of discrete latent embeddings $\mathcal{Z} = \{z_1, \cdots, z_K\}$ (a lookup table with $K = 256$), and used a vector quantization (VQ) loss to learn those embeddings together with InFOM as in VQ-VAE (Van Den Oord et al., 2017). Specifically, given a consecutive transition $(s, a, z, s', a')$, the flow-based intention decoder $q_d(z \mid s, a)$ remains the same, while the intention encoder $p_e(z \mid s', a')$ can now be decomposed into two components: *(1)* the deterministic encoder $p_{\text{enc}} : \mathcal{S} \times \mathcal{A} \to \mathbb{R}^d$ and the quantizer $p_{\text{quant}} : \mathbb{R}^d \to \mathcal{Z}$. The role of the quantizer is to query the closest discrete latent intentions from the encoder outputs using the nearest neighbor,

$$p_{\text{quant}}(p_{\text{enc}}(s', a')) = z_k, \quad \text{where } k = \operatorname{argmin}_{i=1,\cdots,K} \|p_{\text{enc}}(s', a') - z_i\|_2.$$

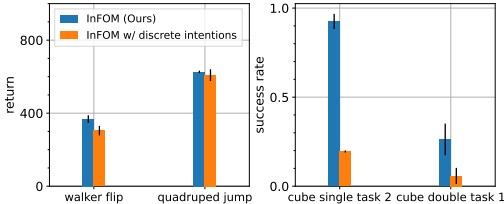

Figure 11: Using discrete intentions slightly decreases InFOM's performance on ExORL tasks ($-11\%$), while drastically decreasing the mean success rate of InFOM on OGBench tasks ($-78\%$).

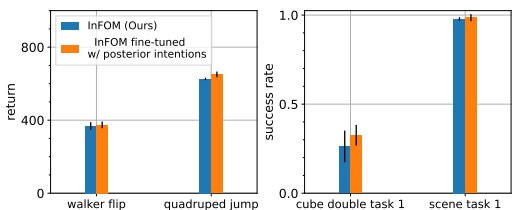

Figure 12: Using the posterior $q_d(z \mid s', a')$ to sample the latents does not significantly change the performance of InFOM ($+7\%$), suggesting that our method is robust against unseen latents. We choose to use the prior $p(z)$ for sampling latents to estimate $Q_z$ throughout our experiments.

Using this quantizer, we replace the surrogate objective in Eq. 4 with the following SARSA flow loss with a vector quantization loss:

$$\mathcal{L}_{\text{SARSA flow}}(p_{\text{enc}}, p_{\text{quant}}, q_d) + \mathcal{L}_{\text{VQ}}(p_{\text{enc}}, p_{\text{quant}}, q_d),$$
$$\mathcal{L}_{\text{VQ}}(p_{\text{enc}}, p_{\text{quant}}, q_d) = \mathbb{E}_{p^\beta(s', a')}[\|\lfloor p_{\text{enc}}(s', a') \rfloor_{\text{sg}} - z_k(s', a')\|_2^2]$$
$$+ \lambda \mathbb{E}_{p^\beta(s', a')}[\|p_{\text{enc}}(s', a') - \lfloor z_k(s', a') \rfloor_{\text{sg}}\|_2^2],$$

where $\lfloor \cdot \rfloor_{\text{sg}}$ denotes the stop gradient operator, and we use straight-through gradients (Bengio et al., 2013) to optimize the SARSA flow loss. During fine-tuning, we use all the discrete latents $\{z_1, \cdots, z_K\}$ to construct intention-conditioned $Q_z$ estimations (Eq. 6) and distill them into the critic $Q$ as in Eq. 7.

We conducted ablation experiments on two ExORL tasks (`walker flip` and `quadruped jump`) and two OGBench tasks (`cube single task 2` and `cube double task 1`) and report performances aggregated over 8 random seeds. Results in Fig. 11 suggest that using discrete intentions slightly decreases InFOM's performance on ExORL tasks ($-11\%$), while drastically decreasing the mean success rate of InFOM on OGBench tasks ($-78\%$). These results indicate that using a continous latent space generally leads to better performance in our experiments.

### F.5 Fine-tuning with posterior intentions

In Sec. 4.4, when estimating the intention-conditioned $Q_z$ for a specific task, we have already sampled the latent $z$ from the prior $p(z)$ instead of the posterior $q_d(z \mid s', a')$. Sampling from the prior, in general, increases the possibility of drawing out-of-distribution latents. We hypothesize that InFOM can generalize over unseen latents on different $(s, a)$ pairs. To quantitatively test this hypothesis, we conduct additional ablation experiments to study the effect of estimating intention-conditioned $Q_z$ using in-distribution latents on the final performance of InFOM. Specifically, we replace the distillation loss in Eq. 7 with a variant that samples $z$ from the posterior $q_d(z \mid s', a')$:

$$\widetilde{\mathcal{L}}(Q) = \mathbb{E}_{(s,a,s',a') \sim p^{\tilde\beta}(s,a,s',a'),\, z \sim q_d(z|s',a')} \left[ L_2^\mu \left( Q_z(s, a) - Q(s, a) \right) \right].$$

We choose to conduct ablation experiments on two ExORL tasks (`walker flip` and `quadruped jump`) and two OGBench tasks (`cube double task 1` and `scene task 1`), aggregating the return and the success rate over 8 random seeds. Results in Fig. 12 indicate that using the posterior to sample the latents for each $Q_z$ does not significantly change the performance of InFOM ($+7\%$). Conversely, these results suggest that InFOM is robust against unseen latents for different $(s, a)$ pairs and using the prior $p(z)$ to sample latents provides sufficient learning signals to drive fine-tuning. We choose to use the prior $p(z)$ for sampling latents to estimate $Q_z$ throughout our experiments.

### F.6 Learning with sparse rewards is challenging

We hypothesize that the sparse reward function on `jaco` tasks explains the performance gap between InFOM and baselines. To test this hypothesis, we conduct ablation experiments on `jaco reach top left` and `jaco reach bottom right`, studying whether using *dense* rewards will mitigate the performance gap. Specifically, the dense reward function is defined as $r(s, g) = -\|s - g\|_2$

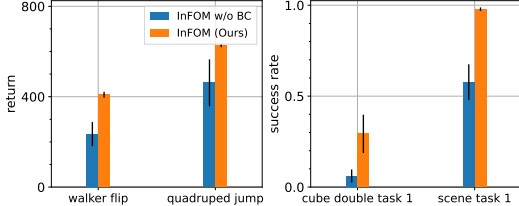
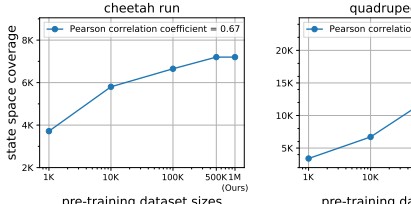

Figure 14: The behavioral cloning regularization in the policy loss is a key component of InFOM.

Figure 15: The diversity of the pre-training datasets has a positive correlation with their sizes.

with $g$ as the target position. To make a fair comparison, we fine-tune the ReBRAC baseline on variants of those two `jaco` tasks with dense reward functions, measuring the performance in the original environments. We report returns across 8 random seeds.

Results in Fig. 13 highlight that using a dense reward function results in $3.6\times$ smaller performance gap, suggesting that the original sparse reward function imposes challenges for learning on `jaco` tasks. We note that Yarats et al. (2022) has also included consistent evidence for this observation, where TD3 + BC (the base algorithm for ReBRAC) performed poorly on the `jaco` domain (Fig. 9 of Yarats et al. (2022)).

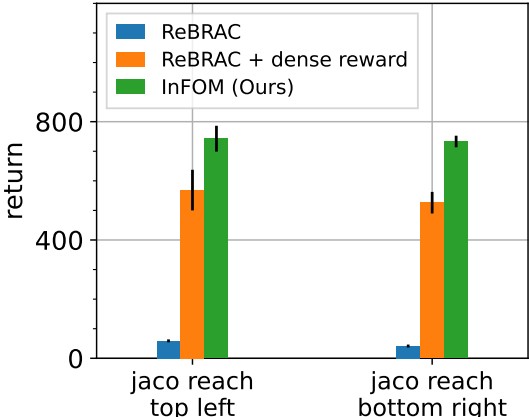

Figure 13: **Reward function structure can impose challenges.** The baseline ReBRAC achieves $3.6\times$ higher performance on variants of `jaco` tasks with a dense reward function.

### F.7 IMPORTANCE OF THE BEHAVIORAL CLONING REGULARIZATION

To study the effect of the BC regularizer (Eq. 8), we conduct experiments comparing a variant of InFOM without the behavioral cloning regularization coefficient ($\alpha = 0$) to our full algorithm with domain-dependent $\alpha$ values (Table 2). We select the same ExORL and OGBench tasks as in Fig. 5 (`walker flip`, `quadruped jump`, `cube double task 1`, and `scene task 1`) and report the means and standard deviations of performance over 8 random seeds after fine-tuning. Results in Fig. 14 suggest that behavioral cloning regularization ($\alpha > 0$) in the policy loss is a key component of our algorithm.

### F.8 DIVERSITY OF THE PRE-TRAINING DATASETS

To quantify the diversity of the pre-training dataset, we conduct a statistical analysis on the datasets for two ExORL tasks (`cheetah run` and `quadruped jump`), analyzing the relationship between the size of the dataset and the diversity of the dataset. Following prior work (Park et al., 2023b), we discretize the continuous state space as a high-dimensional grid (up to $10^{-2}$) and use the number of unique grid points covered by the dataset to measure the diversity. Results in Fig. 15 show that increasing the dataset size induces a higher diversity in the pre-training datasets, with an average correlation coefficient of $0.76$ over those two tasks. Thus, we can study the effect of diverse pre-training datasets on InFOM's performance by varying the pre-training dataset size.

### F.9 THE EFFECT OF DATASET SIZES

**Pre-training dataset size.** Since we aim to predict temporally distant future states from heterogeneous data (Sec. 4.1), InFOM implicitly requires a sufficiently diverse dataset for effective pre-training. To study the relationship between the size of pre-training datasets and the performance of our algorithm, we conduct ablation experiments varying the pre-training dataset size in $\{1K, 10K, 100K, 500K, 1M\}$. We compare the performances of InFOM on two ExORL tasks

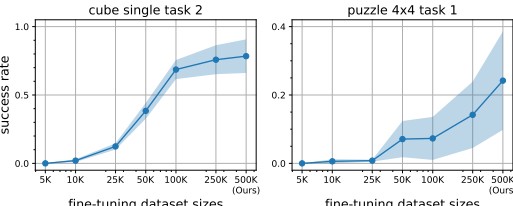

Figure 16: **The effect of pre-training dataset size on InFOM.** Increasing pre-training dataset sizes boosts the final performances of InFOM.

Figure 17: **The effect of fine-tuning dataset size on InFOM.** Increasing the fine-tuning dataset size yields consistent improvements in success rates.

(`cheetah run` and `quadruped jump`) after fine-tuning on the same reward-labeled dataset. We report results across 8 random seeds, following the same evaluation protocol in Appendix D.3.

Results in Fig. 16 indicate that larger pre-training datasets yield higher returns on these tasks. We conjecture that pre-training InFOM on a diverse, reward-free dataset reduces the possibility of sampling out-of-distribution (unseen) intentions, resulting in a higher final performance.

**Fine-tuning dataset size.** We also conduct ablation experiments studying the effect of fine-tuning dataset sizes. Specifically, we select two OGBench tasks (`cube single task 2` and `puzzle 4x4 task 1`) and vary the size of the fine-tuning datasets in $\{5K, 10K, 25K, 50K, 100K, 250K, 500K\}$. Again, we aggregate the performance of InFOM over 8 random seeds, following the same evaluation protocol in Appendix D.3.

Results Fig. 17 show that increasing the fine-tuning dataset size (within the chosen range) yields consistent improvements in success rates on the OGBench tasks. Our explanation for these observations is that the size of the fine-tuning dataset affects the accuracy of the reward prediction.

### F.10 FINE-TUNING ON SUBOPTIMAL DATASETS

We hypothesize that using highly suboptimal fine-tuning datasets will decrease the downstream performance of InFOM. To study the effect of fine-tuning on suboptimal datasets, we conduct ablation experiments on two ExORL tasks (`cheetah run` and `quadruped jump`) because they have dense reward functions and can still produce diverse rewards. To construct suboptimal datasets, we use the reward quantile to filter each transition in the $10^6$ ExORL dataset collected by RND (see Appendix D.1 for details) and then sample $5 \times 10^5$ reward-labeled transitions from the remaining transitions. After constructing these suboptimal datasets, we use them to fine-tune InFOM. Results in Fig. 18 indicate that fine-tuning InFOM on highly suboptimal datasets ($0.2$ reward quantile) achieved only $9\%$ performance of the original InFOM, while using datasets with $0.8$ reward quantile can already achieve $85\%$ performance of the original InFOM. These results suggest that using a sufficiently optimal dataset is important for improving the fine-tuning performance.

### F.11 THE SUFFICIENT NUMBER OF FUTURE STATES IN THE Q ESTIMATION

Since we use MC future states from the InFOM to estimate the intention-conditioned $Q_z$ (Eq. 6), the model may produce unrealistic future states. Thus, the number of future states $N$ affects the accuracy and variance of the Q value estimation (Eq. 6). To investigate the effect of $N$, we conduct ablation studies on a total of 8 tasks, with 4 tasks from the ExORL benchmarks (`cheetah walk`, `walker walk`, `walker flip`, and `quadruped jump`) and 4 tasks from the OG-Bench benchmarks (`cube double task 3`, `puzzle 4x4 task 1`, `cube double task 1`, and `scene task 1`). Below, we report returns and success rates after fine-tuning, aggregating the results over 8 random seeds.

Fig. 19 suggests that, in `cheetah walk` and `puzzle 4x4 task 1`, increasing the number of flow future states yields better performance with consistent variance. In `walker walk` and `cube double task 3`, a larger $N$ does mitigate the high variance in $Q_z$, at the cost of increasing computation. Taken together, these results indicate that a sufficiently large number of flow future states used in $Q_z$ achieves more accurate estimation of Q values, while reducing the variance. In

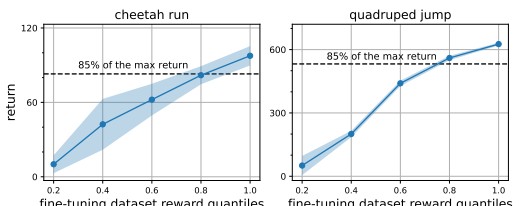
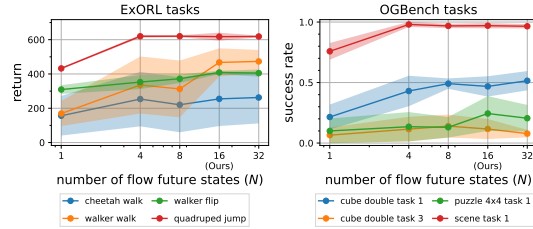

Figure 18: **Fine-tuning on suboptimal datasets.** Fine-tuning on highly suboptimal datasets (0.2 reward quantile) decreased the performance of InFOM, while using a sufficiently optimal (0.8 reward quantile) dataset can already retain the performance.

Figure 19: **Using a sufficient number of flow future states is important.** Increasing the number of flow future states ($N$) in the $Q_z$ estimate boosts the accuracy while reducing variance, resulting in higher final performances of InFOM. We choose $N = 16$ as a balance between the accuracy, variance, and computational constraints in our experiments.

contrast, a smaller number of $N$ potentially yields errors in $Q_z$ from unrealistic future states, resulting in high variance. In practice, our choice of $N = 16$ is a balance between the accuracy, variance, and computational constraints of the estimator.

### F.12 ADDITIONAL HYPERPARAMETER ABLATIONS

We conduct additional ablation experiments on `walker flip`, `quadruped jump`, `cube double task 1`, and `scene task 1` to study the effect of some key hyperparameters in In-FOM (Table 2). Following the same evaluation protocols as in Appendix D.3, we report means and standard deviations across eight random seeds after fine-tuning each variant.

As shown in Fig. 20a, our algorithm is sensitive to the latent intention dimension $d$. Additionally, the effect of the number of steps for the Euler method $T$ (Fig. 20b) saturates after increasing it to a certain threshold ($T = 10$), suggesting the usage of a common value for all tasks.

Results in Fig. 20c, Fig. 20d, and Fig. 20e suggest that the expectile $\mu$ can affect the performance on ExORL tasks, while having minor effects on OGBench tasks. Importantly, the KL divergence regularization coefficient $\lambda$ and the behavioral cloning regularization coefficient $\alpha$ are crucial hyperparameters for InFOM, where domain-specific hyperparameter tuning is required. As discussed in Appendix D.4, we generally select one task from each domain to sweep hyperparameters and then use one set of hyperparameters for every task in that domain.

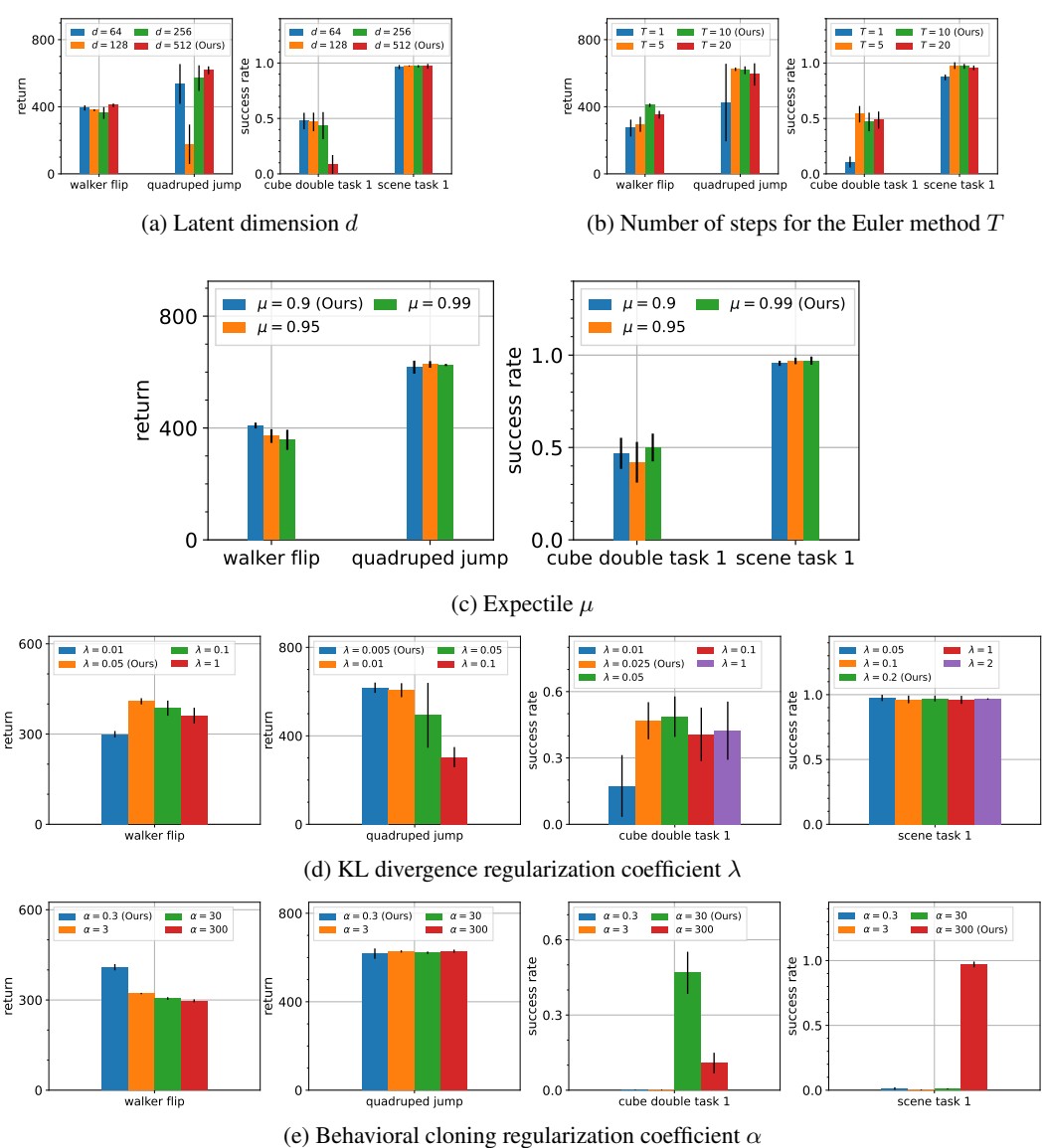

Figure 20: **Hyperparameter ablations.** We conduct ablations to study the effect of key hyperparameters of InFOM as listed in Table 2 on walker flip, quadruped jump, cube double task 1, and scene task 1.

