# OpenReview forum: "Intention-Conditioned Flow Occupancy Models"
_ICLR.cc/2026/Conference — ICLR 2026 Poster_

### Official Review · Reviewer_q22z · 2025-10-25

**Soundness:** 3
**Presentation:** 3
**Contribution:** 3
**Rating:** 8
**Confidence:** 2

**Summary:**

The work provides a way to pre-train a general RL occupancy model using an unlabeled dataset, using a technique close to the newly introduced Temporal-Difference Flows combined with variational inference over the "intention" latent variable, which helps learn the final occupancy model. Given the pretrained flow occupancy model, the authors proposed a way to train a new policy and its Q-value using generalized policy improvement and a Q-value estimate from a reward model and an occupancy measure model. The authors provided extensive experimental validation of their pre- and post-training pipeline across various robotics benchmarks, as well as various ablation studies on the algorithmic choices.

**Strengths:**

- The paper is very well-written and describes the whole training pipeline in great detail.
- The approach bridges the well-developed flow-matching literature with an RL pretraining and subsequent fine-tuning, which distinguishes this approach from prior work on TD-flows.
- The approach of modeling the policy intentions as latent variables seems to be fresh and interesting, since it doesn't require any additional supervision, and distinguishes this approach from Multi-Task RL.
- Strong performance on many benchmarks;

**Weaknesses:**

- The final algorithm combines four (4) neural networks and, at first glance, looks extremely complicated, which can be prone to the accumulation of errors;

**Questions:**

- In Appendix C.1., in the derivation of the ELBO loss, does an inequality (c) (line 1155) require $\lambda \geq 1$ for this derivation?
- Did the newly trained reward, Q-value, and policy models use the features learnt by the occupancy model?
- The improved effect of using the behavior cloning suggests that the dataset consists of high-quality data that is worth being "cloned". What is the issue if the dataset consists only of highly suboptimal but diverse rewards? And how strongly does the diversity of the dataset influence the pre-training performance?
- What are the results on the occupancy measure pretraining performance separately, compared to a standard TD-flows without intention decoding?

---

> ### Author Response · Authors · 2025-11-22
>
> We thank the reviewer for the responses and suggestions for improving the paper. The reviewer brought up three main questions about the paper: (1) the effect of sub-optimal fine-tuning datasets, (2) the effect of diverse pre-training datasets, and (3) a direct comparison of the performance on future state predictions between InFOM and FOM. We have attempted to address these questions through additional ablation experiments and clarifications. **Together with the discussion below, does this fully address the reviewer’s questions?**
>
> > What is the issue if the dataset consists only of highly suboptimal but diverse rewards?
>
> We hypothesize that using highly suboptimal fine-tuning datasets will decrease the downstream performance of InFOM even when the rewards are diverse. To study the effect of suboptimal fine-tuning datasets, we conduct ablation experiments on two ExORL tasks ($\texttt{cheetah run}$ and $\texttt{quadruped jump}$) because they have dense reward functions and can still produce diverse rewards. To construct suboptimal datasets, we use the reward quantile to filter each transition in the 10M ExORL dataset collected by RND (see Appendix D.1 for details) and then sample 500K reward-labeled transitions from the remaining transitions. After constructing these suboptimal datasets, we use them to fine-tune InFOM. Results in Fig. 18 indicate that fine-tuning InFOM on highly suboptimal datasets ($0.2$ reward quantile) achieved only $9\\%$ performance of the original InFOM, while using datasets with $0.8$ reward quantile can already achieve $85\\%$ performance of the original InFOM. These results suggest that using a sufficiently optimal dataset is important for improving the fine-tuning performance.
>
> > And how strongly does the diversity of the dataset influence the pre-training performance?
>
> First, to quantify the diversity of the pre-training dataset, we conduct a statistical analysis on the datasets for two ExORL tasks ($\texttt{cheetah run}$ and $\texttt{quadruped jump}$), plotting the relationship between the size of the dataset and the diversity of the dataset. Following prior work [1], we discretize the continuous state space as a high-dimensional grid (up to $10^{-2}$) and use the number of unique grid points covered by the dataset to measure the diversity. Results in Fig. 15 show that increasing the dataset size induces a higher diversity in the pre-training datasets, with an average correlation coefficient of $0.76$ over those two tasks. Thus, we can study the effect of diverse pre-training datasets on InFOM’s performance by varying the pre-training dataset size.
>
> Second, we conduct ablation experiments studying how the size of the pre-training dataset affects InFOM’s performance after fine-tuning. We use the same two ExORL tasks ($\texttt{cheetah run}$ and $\texttt{quadruped jump}$) and vary the dataset size in $\\{1\mathrm{K}, 10\mathrm{K}, 100\mathrm{K}, 500\mathrm{K}, 1 \mathrm{M} \\}$. Results in Fig. 16 indicate that larger pre-training datasets yield higher returns on these tasks. These results suggest that pre-training InFOM on a diverse, reward-free dataset is important to improving InFOM’s final performance.
>
> > What are the results on the occupancy measure pretraining performance separately, compared to a standard TD-flows without intention decoding?
>
> As mentioned in Sec. 5.2, we included a qualitative visualization showing the latent intentions predicted by InFOM and FOM. We also included additional experiments in Appendix F.2, comparing
> the downstream performance between InFOM and HILP + FOM and FB + FOM after fine-tuning. Nevertheless, we still need additional experiments to investigate the **pre-training** performances of InFOM and FOM. We have revised Appendix F.2 (see the purple text) to include these new experiments.
>
> To quantitatively study the pre-training performances of InFOM and FOM, we conduct ablation experiments comparing the future state predictions from InFOM, HILP + FOM, and FB + FOM on two ExORL tasks ($\texttt{quadruped jump}$ and $\texttt{scene task 1}$). Specifically, we compute the pairwise mean squared error (MSE) between predicted future states and ground-truth future states along a trajectory. We first sample 100 trajectories from the pre-training datasets, and then, for each trajectory, we sample 400 future states from InFOM and the two baselines starting from the same initial $(s, a)$ pair. We compute the pairwise MSE between each sampled future state and the corresponding sequence of ground-truth future states within the same trajectory. The prediction error is reported as the pairwise MSE averaged over all transitions in the 100 trajectories and the 400 sampled future states. Results in Fig. 8 show that InFOM achieves lower prediction errors than the two FOM baselines. Together with Fig. 4, Fig. 6, and Fig. 9, these findings suggest that more accurate flow-based occupancy models benefit both pre-training and fine-tuning on downstream tasks.

---

> ### Author Response · Authors · 2025-11-22
>
> > In Appendix C.1., in the derivation of the ELBO loss, does an inequality (c) (line 1155) require $\lambda \geq 1$ for this derivation?
>
> Correct. We agree that $\lambda \geq 1$ is required for the inequality at line 1209 to hold in our initial submission. In practice, we can use any $\lambda \geq 0$ because rescaling the *input* $(s, a, s_f)$, similar to normalizing the range of images from $\\{0, \cdots, 255\\}$ to $[0, 1]$ in the original VAE [1], preserves the ELBO. Formally, following Higgins et al. (2017) [2], maximizing this ELBO can also be interpreted as an optimization problem that simultaneously predicts future states while penalizing the intention encoder. The $\lambda$ coefficient can then be understood as a Lagrangian multiplier with range $\lambda \geq 0$. We have revised the text (purple) in Sec. 4.2 and Appendix C.1 to include these new clarifications.
>
> > Did the newly trained reward, Q-value, and policy models use the features learnt by the occupancy model?
>
> No, we use separate neural networks to predict the reward $r_\eta$, distill the Q value $Q_\psi$, and train the policy $\pi_\omega$, as in Alg. 2.
>
> We agree that reusing the features (representations) learned by the occupancy models in the reward predictor $r_\eta$, the critic $Q_\psi$, and the policy $\pi_\omega$ is an interesting design decision and might potentially further improve the performance. Since our work considers InFOM as a multi-step generative model instead of as a multi-step representation learning method, we leave the investigation of this design decision as future work.
>
> [1] Kingma et al. (2025). Auto-Encoding Variational Bayes.
>
> [2] Higgins et al. (2017). beta-VAE: Learning Basic Visual Concepts with a Constrained Variational Framework.

---

> > ### Comment · Reviewer_q22z · 2025-11-25
> >
> > I would like to thank the authors for their answer. All my concerns were resolved, and I am happy to keep my positive score.

---

### Official Review · Reviewer_Skrx · 2025-10-27

**Soundness:** 3
**Presentation:** 3
**Contribution:** 3
**Rating:** 6
**Confidence:** 3

**Summary:**

The paper proposes InFOM, a method for unsupervised pre-training in offline reinforcement learning that learns a latent-variable generative model of long-horizon state occupancy conditioned on inferred user intentions. By combining variational inference with flow matching, the authors model the discounted occupancy measure. During fine-tuning, they use implicit generalized policy improvement (GPI) via expectile distillation. The method shows strong empirical gains over prior pre-training approaches.

**Strengths:**

1. Novel integration of flow matching with intention-aware occupancy modeling.

2. Strong empirical results: consistent gains across diverse domains, including challenging sparse-reward and vision-based tasks.

3. Well-motivated design choices, especially the use of SARSA-style bootstrapping and expectile-based implicit GPI to stabilize training.

**Weaknesses:**

1. Choice of Prior over Intentions Lacks Justification. The paper assumes a standard Gaussian prior $p(z)=N(0,I)$ for the latent intention $z$ . While common in VAEs, this choice may be suboptimal for modeling user intentions, which are often discrete or categorical (e.g., “pick”, “place”, “navigate to A”).

Suggestion: The authors should consider discrete latent variables (e.g., via Gumbel-Softmax) to improve interpretability and align better with the semantics of “intentions” in multi-task or goal-directed settings. Ablations comparing continuous vs. discrete priors would strengthen the modeling claim. The current Gaussian prior may encourage over-smoothed representations, potentially conflating distinct behavioral modes (as hinted in Fig. 4, where InFOM already shows better clustering—but could it be sharper with discrete codes?).

2.  Incorrect ELBO Formulation: The parameter $\lambda$ should be at least one, not arbitrary. In equation (3), the coefficient $\lambda$ must satisfy $\lambda \geq 1$ to guarantee that the derived expression is the lower bound.

**Questions:**

Q1: How well does the flow occupancy model generalize to out-of-distribution intentions?

Q2: Is there a risk that the generative occupancy model produces unrealistic futures for novel $z$, harming policy learning?

---

> ### Author Response · Authors · 2025-11-22
>
> We thank the reviewer for helpful suggestions for improving the paper. The reviewer raised three main questions: (1) ablation studies on the choice of the prior over intentions $p(z)$, (2) generalization over out-of-distribution intentions, and (3) the risk of sampling out-of-distribution future states. We have attempted to address (1) and (2) by conducting additional ablation experiments that study discrete $z$ (Appendix F.4) and using posterior $z$ to estimate intention-conditioned $Q_z$ (Appendix F.5). For (3), we provide clarifications using experiments in Appendix F.11 below. **Together with the discussion about other questions, does this address the reviewer’s concerns?**
>
> > Choice of Prior over Intentions Lacks Justification.
>
> We note that the choice of the prior over latent variables $p(z)$ is still an open question in the literature. Prior work has used a standard Gaussian distribution [1], a uniform von Mises–Fisher distribution [2, 3], a continuous uniform distribution [4], and a discrete uniform distribution [5].
>
> As suggested by the reviewer, we ran additional experiments to further investigate the effect of using a discrete set of latent intentions. We have revised the “Additional experiments” paragraph of Sec. 5.3 and Appendix F.4 (see the purple text) to include these new experiments. We selected a set of discrete latent embeddings $\mathcal{Z} = \\{z_1, \cdots, z_K \\}$ (a lookup table with $K = 256$), and used a vector quantization (VQ) loss to learn those embeddings together with InFOM as in VQ-VAE [6]. We defer the detailed formulation to Appendix F.4.
>
> We conducted ablation experiments on two ExORL tasks ($\texttt{walker flip}$ and $\texttt{quadruped jump}$) and two OGBench tasks ($\texttt{cube single task 2}$ and $\texttt{cube double task 1}$) and report performances aggregated over 8 random seeds. Results in Fig. 11 suggest that using discrete intentions slightly decreases InFOM's performance on ExORL tasks ($-11\\%$), while drastically decreasing the mean success rate of InFOM on OGBench tasks ($-78\\%$). These results indicate that using a continous latent space generally leads to better performance in our experiments.
>
> > How well does the flow occupancy model generalize to out-of-distribution intentions?
>
> Below, we describe two forms of generalization, noting that TD learning provides one form of generalization, and we run an additional experiment to study the second type of generalization. We have revised Sec. 4.4 and Appendix F.5 to include these new experiments.
>
> There are two types of generalization over intentions: (1) generalization across trajectories using the same intention, i.e., combinatorial generalization, and (2) generalization over unseen intentions.
>
> For (1), this type of generalization can happen when some transition and intention pairs are never seen together in the same trajectory, but solving the downstream tasks requires stitching together transitions (or intentions) from different trajectory segments in the pre-training datasets. As mentioned in Sec. 4.3, InFOM is capable of combinatorial generalization because we learn it using the SARSA flow matching objective (a temporal difference loss). Appendix B.1 includes further discussions on how InFOM propagates the intention across trajectories.
>
> For (2), we note that when estimating the intention-conditioned $Q_z$ for a specific task, we have already sampled the latent $z$ from the prior $p(z)$ instead of the posterior $q_d(z \mid s', a')$. Sampling from the prior, in general, increases the possibility of drawing out-of-distribution latents. We hypothesize that InFOM can generalize over unseen latents on different $(s, a)$ pairs. To quantitatively test this hypothesis, we conduct additional ablation experiments to study the effect of estimating intention-conditioned $Q_z$ using in-distribution latents on the final performance of InFOM. Specifically, we replace the distillation loss in Eq. 7 with a variant that samples $z$ from the posterior $q_d(z \mid s', a')$:
> \begin{align*}
>     \widetilde{\mathcal{L}}(Q) = \mathbb{E}_{(s, a, s’, a’) \sim p^{\tilde{\beta}}(s, a, s’, a’), \: z \sim q_d(z \mid s’, a’) } \left[ L_2^{\mu} \left(Q_z(s, a) -  Q(s, a) \right) \right].
> \end{align*}
> We choose to conduct ablation experiments on two ExORL tasks ($\texttt{walker flip}$ and $\texttt{quadruped jump}$) and two OGBench tasks ($\texttt{cube double task 1}$ and $\texttt{scene task 1}$), aggregating the return and the success rate over 8 random seeds. Results in Fig. 12 indicate that using the posterior to sample the latents for each $Q_z$ does not significantly change the performance of InFOM ($+7\\%$). Conversely, these results suggest that InFOM is robust against unseen latents for different $(s, a)$ pairs and using the prior $p(z)$ to sample latents provides sufficient learning signals to drive fine-tuning. We choose to use the prior $p(z)$ for sampling latents to estimate $Q_z$ throughout our experiments.

---

> ### Author Response · Authors · 2025-11-22
>
> > Is there a risk that the generative occupancy model produces unrealistic futures for novel $z$, harming policy learning?
>
> Yes. We agree that InFOM might sample unrealistic future states for a fixed $z$: there might be errors in the numerical solver. Note that we choose to sample $z$ from the prior $p(z)$ instead of from the posterior $q_d(z \mid s', a')$, resembling drawing random samples from a VAE [7]. Sampling from the prior $p(z)$ will increase the likelihood of drawing “novel” z as mentioned by the reviewer.
>
> Because sampling from the InFOM might produce unrealistic future states for a fixed $z$, we sample multiple future states ($N$) to estimate the latent-conditioned Q estimation $Q_z$ as in Eq. 6. Importantly, the choice of the number of future states $N$ affects the accuracy and variance of our Q estimate. In Appendix F.11, we include ablation experiments studying the effect of $N$ on the downstream performances of InFOM on **8** tasks. We report returns and success rates after fine-tuning, aggregating the results over 8 random seeds. Results in Fig. 19 suggest that a smaller number of $N$ potentially yields larger errors in $Q_z$ from unrealistic future states, resulting in high variance. In contrast, a sufficiently large number of future states used in $Q_z$ achieves more accurate estimates of Q values, while reducing the variance. In practice, our choice of $N = 16$ is a balance between the accuracy, variance, and computational constraints of the estimator. We have revised the text (purple) in Appendix F.11 to highlight this risk.
>
> > The parameter $\lambda$ should be at least one, not arbitrary. In equation (3), the coefficient $\lambda$ must satisfy $\lambda \geq 1$ to guarantee that the derived expression is the lower bound.
>
> We agree that $\lambda \geq 1$ is required for the inequality at line 1209 to hold in our initial submission. In practice, we can use any $\lambda \geq 0$ because rescaling the *input* $(s, a, s_f)$, similar to normalizing the range of images from $\\{0, \cdots, 255\\}$ to $[0, 1]$ in the original VAE [7], preserves the ELBO. Formally, following Higgins et al. (2017) [8],
> maximizing this ELBO can also be interpreted as an optimization problem that simultaneously
> predicts future states while penalizing the intention encoder. The $\lambda$ coefficient can then be understood as a Lagrangian multiplier with range $\lambda \geq 0$. We have revised the text (purple) in Sec. 4.2 and Appendix C.1 to include these new clarifications.
>
>
> [1] Kevin et al. (2024). Unsupervised Zero-Shot Reinforcement Learning via Functional Rewa
>
> [2] Park et al. (2024). Foundation Policies with Hilbert Representations.
>
> [3] Touati et al. (2021). Learning One Representation to Optimize All Rewards.
>
> [4] Sharma et al. (2019). Dynamics-Aware Unsupervised Discovery of Skills.
>
> [5] Eysenbach et al. (2018). Diversity is All You Need: Learning Skills without a Reward Function.
>
> [6] Oord et al. (2017). Neural Discrete Representation Learning.
>
> [7] Kingma et al. (2025). Auto-Encoding Variational Bayes.
>
> [8] Higgins et al. (2017). beta-VAE: Learning Basic Visual Concepts with a Constrained Variational Framework.

---

> > ### Comment · Reviewer_Skrx · 2025-11-27
> >
> > Dear authors, thank you for the informative rebuttal. I am ready to raise my score by 1 point and continue to support acceptance of the paper.

---

### Official Review · Reviewer_Eojh · 2025-11-05

**Soundness:** 4
**Presentation:** 4
**Contribution:** 4
**Rating:** 8
**Confidence:** 2

**Summary:**

This work leverages recent advances in generative models in the direction of flow-matching algorithms in order to tackle the paradigm of pre-training and finetuning RL models. This is done by learning occupancy models over the state space using flow matching (as done in Farebrother et al. (2025)) while  taking into account the fact that large pretraining datasets usually contain a mixture of intentions since they are collected by different users. The proposed approach InFOM explicitly models the intention as a latent variable using a VAE which is used to condition the occupancy model. While fine-tuning, the method generates intention-conditioned Monte-Carlo estimates of  the crictic from sampled future states and then distills them into a single critic via an upper-expectile loss. The authors show that across 36 state-based and 4 image-based tasks ), InFOM matches or outperforms strong pre-train-and-fine-tune baselines, reporting a 1.8× median return gain and a 36% success-rate increase, with particularly large gains on harder manipulation domains.

**Strengths:**

* The paper is well written and easy to follow
* The experimental results showing the proposed approach outperforms the baselines in most of the case with the gap widening on more difficult tasks
* A large number of baselines have been included and sufficient experimental detail is provided.
* I really the like the qualitative analysis of the learnt latent intention model.

**Weaknesses:**

The proposed approach builds on top of Farebrother et al. (2025) which uses flow matching to learn occupancy models, by explicitly modeling intention as a latent variable. Including an ablation / baseline comparing the downstream performance with intention conditioning of the occupancy model vs without conditioning it seems to be missing.

**Questions:**

Is there prior work one using other generative modeling approaches, specifically diffusion models for learning occupancy models? If yes, could the authors provide some intuition on comparison between using diffusion versus flow matching for learning occupancy models?

---

> ### Author Response · Authors · 2025-11-22
>
> We thank the reviewer for the constructive feedback and suggestions. The reviewer raises two main questions: (1) ablation studies comparing the downstream performance between InFOM and FOM (flow occupancy models pre-trained without conditioning on intentions) and (2) alternative generative methods for modeling occupancy measures. To compare the policy performance between InFOM and FOM, we include ablation studies in Appendix F.2 (original submission). Below, we answer the question about alternative methods for modeling occupancy measures by mentioning discussions and experiments in prior work [1]. **Together with our discussions below, do these address the reviewer’s concerns?**
>
> > Including an ablation / baseline comparing the downstream performance with intention conditioning of the occupancy model vs without conditioning it seems to be missing.
>
> Thanks for the suggestion. We have already included ablation studies comparing the policy performance between InFOM and FOM in Appendix F.2 (If we have misunderstood the question, we’d be happy to run additional experiments.). In these experiments, we replace the variational intention encoder in InFOM with two prior unsupervised representation learning methods HILP [2] and FB [3], and then pre-train flow occupancy models (not conditioning on intentions) on top of them. Note that FB + FOM is equivalent to TD flows with GPI in Farebrother et al. (2025) [1]. Results in Fig. 9 indicate that InFOM can outperform HILP + FOM and FB + FOM on 3 of 4
> tasks, while being simpler. We have revised the text (purple) at the end of Sec. 5.2 to emphasize these experiments.
>
> > Is there prior work one using other generative modeling approaches, specifically diffusion models for learning occupancy models? If yes, could the authors provide some intuition on comparison between using diffusion versus flow matching for learning occupancy models?
>
> Yes, the paper mentioned by the reviewer, Farebrother et al. (2025) [1], has discussed using alternative prior generative modeling approaches to learn the occupancy models. We have also compared with Farebrother in Fig. 8 and Fig. 9. Specifically, Farebrother et al. (2025) compare flow-based occupancy models against representative methods, including denoising diffusion [4], VAE [5], and GAN [6]. Results in Fig. 2 of Farebrother et al. (2025) [1] show that TD flow (TD$^2$-CFM in the figure) has already outperformed alternative generative methods in modeling the occupancy measures. For this reason, we do not include comparisons against alternative generative occupancy models to distinguish our contributions. Nevertheless, we have included an additional paragraph (see the purple text) at the end of Sec. 5.3 to mention these comparisons.
>
>
> [1] Farebrother et al. (2025). Temporal Difference Flows.
>
> [2] Park et al. (2024). Foundation Policies with Hilbert Representations.
>
> [3] Touati et al. (2021). Learning One Representation to Optimize All Rewards.
>
> [4] Ho et al. (2020). Denoising Diffusion Probabilistic Models.
>
> [5] Kingma et al. (2013). Auto-Encoding Variational Bayes.
>
> [6] Goodfellow et al. (2014). Generative Adversarial Networks.

---

> > ### Comment · Reviewer_Eojh · 2025-11-26
> >
> > I would like to thank the authors for the informative response. It addresses my concerns. However, I would like to maintain my  score.

---

### Official Review · Reviewer_DcLe · 2025-11-12

**Soundness:** 3
**Presentation:** 1
**Contribution:** 3
**Rating:** 2
**Confidence:** 2

**Summary:**

The present work leverages flow-matching models to predict trajectories of future states for RL tasks to help the actor finding better policies.
Extensive experiments shows on gym and robotic manipulation benchmarks  shows promissing results.

**Strengths:**

- The idea of leveraging "intent" for multi-step generation especially in the multi-agent setting or iterative refininement with human input.
- Extensive experiments show practicability of the method.

**Weaknesses:**

Major

 After several reading, I fail to understand how the algorithm works. It should not be that hard to understand:
- flow-matching to predict future states is straightforward given trajectories either conditionned or guided on observed states.
- the predicted latent variable has to be used somehow by the policy, either by boosting an pre-trained one or by training from scratch.

 However, the main body of the paper discusses Q-functions which in the end are conditioned on the intention. This leads the reader to infer that the intention variable is infered and fed to the agent along actor trajectories.

My wild guess: since the policy is mainly trained via a behaviorial cloning loss (see algorithm 2 in appendix) which is barely mentioned in the main body,  using the predicted state actions, the flow-matching is merely doing data-augmentation on the off-line dataset.

I thus think that the abstract is misleading, sections 2 and 3 should be rewritten entirely to focus method rather than the intention (ironically). Algorithm 2 is rather involved and should be explained fully in the main body of the paper, all losses should be introduced clearly.


Minor

1- Diffentiating through an ODE solver is associated to a citation to Park et al. 2025b.  However, this is a problem already accounted for in the Normalizing flow litterature, see FFJORD https://arxiv.org/abs/1810.01367 or even Neural ODE https://arxiv.org/abs/1806.07366. Considering that Ricky Chen is a major contributor to both flow matching and Normalizing flow literature, it is odd to cite Park et al for this aspect.

2- "The deterministic nature of ODEs equips flow-matching methods with simpler learning objectives and faster inference speed than denoising diffusion models (Lipman et al., 2023; 2024; Park et al., 2025b)" is problematic for two reasons. I doubt that the training objective is simpler for flow-matching compared to Diffusion model if by that the authors mean the target function to learn or even the numerical stability of the loss. It is true however that the numerical stability of the *inference* is better for FM compared to DDPM. It is also true that the MSE-based loss of FM is more numerically stable and lighter than the KL-based loss of ODE-based Normalizing flows   such as FFJORD, see for instance https://arxiv.org/abs/2107.07232. The second reasin is that Park et al. 2025b has little to do with the statement.

**Questions:**

1- How does the inference work ? No algorithm is provided.

2- How do you intend to use this method for pre-trained policies ?

---

> ### Author Response · Authors · 2025-11-22
>
> We thank the reviewer for the careful review. It seems like there’s a bit of confusion about how exactly the method works (e.g., what are the inputs and outputs to various components). We’ll try to explain the details below, and we will work hard to revise the paper to make sure that these details are very clear in the revised text.
>
> So, how does the method work? In broad strokes, the reviewer is right that we’re learning an occupancy measure, and then using that occupancy measure to update the policy. However, the way the policy is updated is different from how the reviewer described. The policy just takes as input the state (not the intention) and produces the action. The objective for learning the policy is Eq. 8. The intuition is that we want to select the action that maximizes the following:
> \begin{align*}
> \pi(a \mid s) \gets \text{argmax}_a \max_z Q_z(s, a).
> \end{align*}
> We choose to distill the inner maximization $\max_z Q_z(s, a)$ into a single critic $Q(s, a)$ (Eq. 7). Therefore, for training the policy, we don’t need to infer the $z$.
>
> For training the occupancy measure, we do need to infer the z, as the training data is not labeled with $z$. We do this in a manner similar to a VAE, which likewise is fit to a dataset that is not labeled with $z$ [1].
>
> **Does this clarify how the method works?**
>
> We’ve gone ahead and revised the paper to clarify this point (see the purple text in Sec. 4.4, Alg. 1, and Alg. 2). We thank the reviewer again for raising this question about the method; revising the paper to clarify this point will ensure that the paper is as readable as possible. Below, we’ll address other questions raised in the review.
>
> >  How does the inference work?
>
> The inference of $z$ happens in both pre-training and fine-tuning. During pre-training, we infer $z$ as the latent intentions of unlabeled data by using an ELBO similar to VAE (Eq. 3). We learn a variational encoder to encode the latent $z$ and use a flow-based model to learn the decoder, which predicts the future state.
>
> During fine-tuning, we infer the task-specific latent variable $z$ by GPI. Introduced in Sec. 4.1 in Barreto et al. (2017) [2], GPI is a widely used technique to select one Q estimation from a finite *set* of Q estimations for different policies: simply selecting the Q estimation with the highest value and then using it to learn the policy. Theorem 1 of Barreto et al. (2017) also provides a theoretical guarantee that the resulting policy from GPI will have a higher Q than any other policies forming the set of Q estimations. In our scenario, the set of Q estimations corresponds to the Monte Carlo $Q_z$ for different latents $z$s (Eq. 6). GPI will first select the Q with the highest value
> \begin{align*}
> \max_{z^{(1)}, \cdots, z^{(M)}: z^(j) \sim p(z)} Q_{z^{(j)}}(s, a),
> \end{align*}
> and then use it to update the policy
> \begin{align*}
> \text{argmax}_{\pi} \mathbb{E} _{ a \sim \pi(a \mid s) }  \left[ \text{max} _\{ z^{(j)} \} Q _{z^{(j)}}(s, a) \right].
> \end{align*}
> We have combined these two steps into a single equation between lines 301 and 303. We omit the details of these two steps because they have been thoroughly discussed in Barreto et al. (2017) and are not our contribution.
>
> > How do you intend to use this method for pre-trained policies?
>
> Our method works both in settings when we’re given pre-trained policies (line 12 in Alg. 1) and when we’re not given pre-trained policies. In settings where we are given a pre-trained policy or we pre-trained a behavioral cloning policy, we use it to initialize the policy for fine-tuning. We note that this step is entirely optional. In settings where we are not given a pre-trained policy, we initialize the fine-tuning policy from random initialization and train it from scratch using the loss in Eq. 8.
>
> > since the policy is mainly trained via a behaviorial cloning loss (see algorithm 2 in appendix) which is barely mentioned in the main body
>
> Our policy is learned by maximizing the distilled critic (Eq. 7) while minimizing a behavioral cloning regularization as in Eq. 8. As mentioned in lines 317 and 318, we included this behavioral cloning regularization term in Eq. 8 to prevent sampling OOD actions and mitigate error propagation through $Q_z$. We have revised Alg. 2 to highlight the corresponding equations.
>
> In Appendix F.7, we have already included an ablation study investigating the effect of this behavioral cloning regularization. Results in Fig. 14 show that InFOM without a behavioral cloning regularization ($\alpha = 0$) can already achieve non-trivial downstream performance, while adding the domain-dependent $\alpha > 0$ values can boost performance. We also ablate the value of $\alpha$ on some tasks in Fig. 20 (e). Nevertheless, we have revised the text after Eq. 8 (purple) to emphasize these ablation experiments.

---

> > ### Author Response · Authors · 2025-11-22
> >
> > > Algorithm 2 is rather involved and should be explained fully in the main body of the paper, all losses should be introduced clearly.
> >
> > As mentioned in the “Algorithm summary” paragraph of Sec. 4.4, Alg. 2 mainly includes three components: (1) learning a reward predictor $r_{\eta}$ using regression for estimating $Q_z$ in Eq. 6, (2) distilling the critic $Q_{\psi}$ (not conditioned on the latent $z$) using Eq. 7, and (3) extracting the policy by maximizing the $Q_{\psi}$ while incorporating behavioral cloning regularization in Eq. 8.
> >
> > Specifically, line 8 in Alg. 2 correspond to the reward regression, line 10 in Alg. 2 corresponds to $Q_z$ estimation in Eq. 6, line 11 in Alg. 2 corresponds to the critic distillation in Eq. 7, and line 12 in Alg. 2 corresponds to the policy extraction in Eq. 8. In Alg. 1 and Alg. 2, we have added the equation numbers (purple) for clear references.
> >
> > > Diffentiating through an ODE solver is associated to a citation …
> >
> > Thanks for the suggestion. We have revised the citations at the end of Appendix A.2 to include the papers mentioned by the reviewer.
> >
> > > "The deterministic nature of ODEs equips flow-matching methods with simpler learning objectives and faster inference speed than denoising diffusion models (Lipman et al., 2023; 2024; Park et al., 2025b)" is problematic for two reasons …
> >
> > Thanks for the suggestion. We have revised the citation and replaced the “simpler” with “more stable” in this sentence. In Lipman et al. (2022) [3], the authors indeed mentioned that flow matching achieved faster training and sampling than diffusion models, so we preserve the word “faster”.
> >
> > To summarize, we have attempted to clarify the overall framework of our algorithm, answering specific questions from the reviewers, and revised misleading sentences. We are happy to further explain our algorithm and incorporate suggestions if the reviewer finds any component in the paper difficult to understand.
> >
> > [1] Kingma et al. (2013). Auto-Encoding Variational Bayes.
> >
> > [2] Barreto et al. (2017). Successor features for transfer in reinforcement learning.
> >
> > [3] Lipman et al. (2022). Flow Matching for Generative Modeling.

---

> ### Comment · Reviewer_DcLe · 2025-11-25
>
> I thank the authors for their clarifications.
> I still find the presentation is the paper imperfectly clear.  I don't feel comfortable reproducing the experiments myself.
> I still think that discussing the inference algorithm in the main text would be helpful compared to the current writing that spend too much time to my taste on a rather "hand-wavy" motivational discussion without formal statement (like a theorem). Although, there is no need for such formal statements given my understanding of the work.
>
> I am raising my score to weak accept to take into account the significative body of ablation that is included. I however suggest the author to continue iterating on the presentation.

---

### Author Response · Authors · 2025-12-02
**Discussion summary**

Dear AC,

We sincerely appreciate your time and effort in evaluating this work during this exceptional situation. In case it's helpful, we would like to provide a summary of the discussions so far.

We have worked hard to incorporate the reviewers' feedback with additional experiments and revisions. Before the reviews were rolled back, we received replies from all reviewers confirming that we addressed their concerns. The most recent ratings (as of Nov 27) were **6, 8, 8, 8** from reviewers DcLe, Eojh, Skrx, and q22z, respectively. Reviewers Eojh and q22z maintained their initial positive ratings (8). Reviewer Skrx explicitly mentioned raising the rating from **6 to 8** to continue supporting the acceptance of the paper (see [revision history](https://openreview.net/revisions?id=gv4AZVywaW)).

Among all the reviewers, reviewer DcLe is the only one who initially gave a negative rating of 2, as of Nov 25 (two days before the incident). In the initial review, reviewer DcLe's main concern is about how exactly the method works, e.g., what are the inputs and outputs to various components. We have explained the generalized policy improvement procedure, the policy update strategy, and the role of the behavioral clone regularization to clarify the confusion (see [our](https://openreview.net/forum?id=j6D83Mf6LG&noteId=bVckoSHIF9) [responses](https://openreview.net/forum?id=j6D83Mf6LG&noteId=aLlttzjrJ1)). We've gone ahead and revised the paper to include these clarifications (see the purple text in Sec. 4.4, Alg. 1, and Alg. 2). Upon reading our responses, reviewer DcLe explicitly [mentioned](https://openreview.net/forum?id=j6D83Mf6LG&noteId=OBoKkwNu99) raising the rating from **2 to 6** to take into account the new ablation experiments and improved presentation (see [revision history](https://openreview.net/revisions?id=VHNOk3hSQL)).

We have included new experimental results in the revised paper and incorporated specific writing suggestions mentioned by the reviewers. We hope that this summary helps assess our submission. Please feel free to let us know if there are any other concerns or questions. Thank you again for your time and effort in reviewing our work under these unusual circumstances.

The Authors

---

### Meta-Review · Area_Chair_xQAk · 2026-01-07

**Summary:**

The paper proposes InFOM, a pre-train + fine-tune framework for RL that learns a long-horizon occupancy-measure based generative model using flow matching, while treating “intention” as a latent variable because large offline pretraining datasets mix behaviors from different users/policies. Empirically, it reports strong gains across a broad set of state-based and image-based RL benchmarks, with larger gains on harder manipulation tasks.

All reviewers recognized the contribution of this paper and recommended to accept this paper.

**Reviewer Concerns:**

Most of the reviewers' concerns are address and acknoeldged by the reviewers.

**Reviewer Scores:**

One reviewer has reasonably acknowledge to raise the score from 2 to 5, other reviewers already hold positive enough score.

---

### Decision · Program_Chairs · 2026-01-26

Accept (Poster)